# Positive Mental Health Questionnaire (PMHQ) for Healthcare Workers: A Psychometric Evaluation

**DOI:** 10.3390/healthcare11233041

**Published:** 2023-11-26

**Authors:** Juan José Luis Sienra-Monge, David Luna, Rosa Paola Figuerola-Escoto, Itzihuari Iratzi Montufar-Burgos, Alejandra Hernández-Roque, Arturo Soria-Magaña, Filiberto Toledano-Toledano

**Affiliations:** 1Unidad de Pediatría Ambulatoria, Hospital Infantil de México Federico Gómez, National Institute of Health, Dr. Márquez 162, Doctores, Cuauhtémoc, Mexico City 06720, Mexico; jjsienra@hotmail.com (J.J.L.S.-M.); psicalehr@gmail.com (A.H.-R.); asoria@himfg.edu.mx (A.S.-M.); 2Unidad de Investigación Multidisciplinaria en Salud, Instituto Nacional de Rehabilitación Luis Guillermo Ibarra Ibarra, Calzada México-Xochimilco 289, Arenal de Guadalupe, Tlalpan, Mexico City 14389, Mexico; xeurop@hotmail.com; 3Sección de Estudios de Posgrado e Investigación, Centro Interdisciplinario de Ciencias de la Salud Unidad Santo Tomás, Instituto Politécnico Nacional, Av. de los Maestros s/n, Santo Tomás, Miguel Hidalgo, Mexico City 11340, Mexico; rfiguerolae@ipn.mx; 4Facultad de Psicología, Universidad Nacional Autónoma de México, Av. Universidad 3004, Ciudad Universitaria, Mexico City 04510, Mexico; ira.montufar@comunidad.unam.mx; 5Unidad de Investigación en Medicina Basada en Evidencias, Hospital Infantil de México Federico Gómez, Instituto Nacional de Salud, Dr. Márquez 162, Doctores, Cuauhtémoc, Mexico City 06720, Mexico; 6Dirección de Investigación y Diseminación del Conocimiento, Instituto Nacional de Ciencias e Innovación para la Formación de Comunidad Científica, INDEHUS, Periférico Sur 4860, Arenal de Guadalupe, Tlalpan, Mexico City 14389, Mexico

**Keywords:** positive mental health, psychometric, confirmatory factor analysis, internal consistency, assessment, SEM

## Abstract

The Positive Mental Health Questionnaire (PMHQ) has been validated across various populations but has displayed diverse psychometric structures depending on the procedures used. The original version of the PMHQ includes 39 items organized into 6 factors, although there are reports that indicate a reduced structure of between 1 and 4 factors. The aim of this study was to assess the psychometric properties of the PMHQ with 1, 4 and 6 factors. A total of 360 healthcare workers aged 23 to 77 (M = 37.06; SD = 10.79) participated. Construct validity was assessed through confirmatory factor analysis using weighted root mean square residual. The original 6-factor (χ^2^/df: 3.40; RMSEA: 0.085; CFI: 0.913; TLI: 0.906) and a reduced 4-factor (χ^2^/df: 2.90; RMSEA: 0.072; CFI: 0.931; TLI: 0.926) structure showed acceptable fit. The fit of the 1-factor model was unacceptable. The internal consistency was evaluated through McDonald’s ω, and it was acceptable for 4 of 6 factors of the original structure and for 3 of 4 factors of the reduced structure. In conclusion, these findings suggest that the 6-factor and 4-factor models are valid for measuring positive mental health. However, issues with internal consistency must be investigated.

## 1. Introduction

### 1.1. Positive Mental Health

Given the recent recognition of the importance of mental health, in 2019, the WHO launched an initiative that guaranteed access to quality care for more than 100 million people in 12 priority countries. However, the incidence of people with mental health disorders is increasing, and treating the most common conditions costs the global economy US$1 trillion annually [1]. In this regard, focusing on the salutogenesis principles, which places positive mental health (SMP) in well-being, rather than in the absence of disease [2,3,4], leads to cost reduction and better prognosis if the disease manifests itself, since the person who suffers from it will focus on positive emotions, commitment, purpose, positive relationships and achievements [4,5], recognizing the environment and the community as facilitating factors and adapting based on an optimistic and problem-solving perspective [6] in favor of psychological resilience [7]. Thus, SMP is associated with a reduction in the predisposition to mental illnesses [8], with a strengthening of resilience [9], and with greater self-care behaviors in health [10] and has even been negatively associated with suicidal ideation [11]. Logistic regression analyses have also shown that SMP positively influences recovery from anxiety disorders [12] and that the absence of SMP increases the likelihood of mortality from known causes for men and women of all ages [13].

### 1.2. Positive Mental Health in Healthcare Workers

The recent pandemic caused by coronavirus disease 2019 (COVID-19) has focused our attention on the emotional stability of healthcare workers. Based on the report of a new pneumonia in 2019, which emerged in Wuhan, China, healthcare workers on the front lines of care showed a very high prevalence of mental health disorders. Then, various studies on the prevalence of the different disorders identified on the Asian continent emerged; in this regard, we can cite that of Lai et al. [14], in which high rates of depression, anxiety, insomnia and anguish were reported (50.4%, 44.6%, 34.0% and 71.5%, respectively) in the hospital community. Kang et al. [15] noted that 28.6% of healthcare workers had moderate and severe disorders. In Western countries, very significant percentages of high stress (71.1%) were also detected among anesthesiologists, in addition to symptoms of insomnia in 36.7%, anxiety in 27.8% and depression in 51.1% [16]. Among UK physicians, Johns et al. [17] reported a considerable prevalence of anxiety (26.3%), depression (21.9%), posttraumatic stress (11.8%) and burnout (10.8%), which increased among those who showed low psychological flexibility and intolerance to uncertainty. In another study among doctors from Bangladesh, the data coincided with a high prevalence of depression and anxiety, 67.72% and 48.5%, respectively [18].

From a salutogenic perspective, there have been few studies among healthcare workers. We can cite, for example, a study that concluded by noting the importance of psychosocial well-being in healthcare workers, finding an important association between job satisfaction and positive mental health [19]. Further research revealed that healthcare workers with flourishing mental health are less likely to receive a diagnosis of posttraumatic stress disorder [20]. Dyrbye et al. [21] reported that an improvement in mental health in medical students decreased the prevalence of suicidal ideation, thoughts of desertion and the presence of unprofessional behavior and generated an improvement in altruistic beliefs related to the responsibility of doctors in society. Finally, Bajo et al. [22] noted that one way to improve the desired social well-being in the population of physicians is social recognition, which is often absent.

In general, although the affectation caused by the workload among healthcare workers is evident, the protection of their emotional stability is insufficient and scarcely reported. In relation to this, a systematic review of publications that analyzed the functionality of SMP interventions for the prevention of mental illnesses in health personnel was carried out and found that only 4 published articles had sufficient quality; it was concluded that research has mostly concentrated on stressors and not on improving well-being or developing positive mental health [23].

### 1.3. Self-Report Instruments to Assess Positive Mental Health

The first instrument to be analyzed was the Lluch Positive Mental Health Questionnaire (PMHQ) [24], which we will discuss in detail later, since the present study sought its validation in healthcare workers in Mexico. The PMHQ has 39 items structured into 6 factors that correspond to the 6 positive mental health criteria of Jahoda [25], who conceives of mental health from an individual perspective. Although she accepts the impact of the environment and culture on health and illness, she refuses to speak of “sick societies” because although this concept fully accepts the mutual influence of the physical and mental aspects of the human being, it does not consider that physical health is sufficient for good mental health. Another highly relevant instrument is the Mental Health Continuum-Large Form (MHC-LF) with 40 items, 7 focused on evaluating emotional well-being, 18 on psychological well-being and 15 on social well-being, with adequate estimates of internal consistency. This instrument conceives a model that places health and illness as correlated unipolar dimensions that form a complete state of mental health, not a single bipolar dimension and that places mental health as the simple absence of psychopathology; likewise, the model on which it is based presumes that mental health is the ultimate goal of personal and social functioning [26]. Derived from the Mental Health Continuum-Large Form, the Mental Health Continuum-Short Form (MHC-SF) retains the 14 most prototypical items of the long version. Further, the MHC-SF points out that positive mental health is not simply the absence of mental illness but includes the presence of positive feelings (emotional well-being) and positive functioning in individual life (psychological well-being) and in community life (social well-being). This brief version assesses the frequency with which respondents experience positive mental health, with a score ranging from 0 (none of the time) to 5 (all the time), thus enabling their classification; for example, to have flourishing mental health, they must experience every or nearly every day at least one of the 3 signs of hedonic well-being and at least 6 of the 11 signs of positive functioning during the past month. Regarding its psychometric properties, the MH-SF has shown excellent internal consistency and discriminant validity in adolescents and adults from the United States, the Netherlands and South Africa [27,28,29,30,31]. Another relevant instrument is the General Health Questionnaire (GHQ-12), which is a screening instrument that detects nonpsychotic psychiatric disorders [32]; it has been translated and adapted into multiple languages, as it has the advantage of brevity and has shown adequate psychometric properties, which is why it is considered the most widely used screening instrument worldwide [33]. However, the disadvantage is a highly debated structure that some studies have indicated examines two factors: depression/anxiety and social dysfunction [34,35,36,37,38,39]. Other studies describe three factors: coping strategies, self-esteem and stress [40,41,42,43], and there are authors who recommend using the GHQ-12 as a unidimensional screening instrument [44,45,46]. Another instrument is the Positive Mental Health (PMH) instrument, which evaluates six factors: general coping, personal growth and autonomy, spirituality, interpersonal skills, emotional support and global affect; it consists of 47 items and was developed and validated with participants from China, Malaya and India, showing high internal consistency and correlation with other measures of well-being [47]. A brief version was prepared from the PHQ of 47 items, and the 19-item Positive Mental Health (PMH-19), validated with populations from Singapore, China, Malaya and India, also showed high internal consistency [48]. In addition, the Achutha Menon Centre Positive Mental Health Scale (AMCPMHS) developed for the Indian population is a valid and reliable instrument of 20 items organized into 4 dimensions: (1) realization of one’s own potential and belief in the dignity and worth of self, (2) utilization of coping abilities, (3) belief in the worth of others and (4) work productivity and community contribution [49]. Lukat et al. [50] also developed a 9-item unidimensional positive mental health scale (PMH Scale) and validated it in various relevant groups. We can highlight the validation with parents of children with cancer, where it proved to be a valid, reliable and culturally relevant scale [51]. Finally, the Rapid Positive Mental Health Instrument (R-PMHI) is a 6-item unidimensional scale developed and validated with the Singaporean population [52]. In terms of validation with the Mexican population, we can highlight the Positive Mental Health Scale for Adults as relevant, which includes 7 factors and 83 items and has a global internal consistency α = 0.962 that ranges from 0.644 to 0.954 between factors [53].

### 1.4. PMHQ Validation Studies

To evaluate positive mental health objectively and from a multifactorial model, Llinch developed the Positive Mental Health Questionnaire (PMHQ) based on the Jahoda model [25], which he validated through an exploratory factor analysis (EFA) among nursing students from the University of Barcelona. In this regard, factor analysis is a method that allows modeling the covariation between a set of observed variables based on a latent construct, and through EFA, the generation of hypotheses about its possible structure [54]; thus, in this exploration, Llinch detected the presence of six factors: personal satisfaction, prosocial attitude, self-control, autonomy, problem-solving and self-actualization and interpersonal relationship skills. Llinch obtained a global internal consistency α = 0.90 ranging from 0.58 to 0.82 between factors, with 46.8% of the variance explained [24]. After its development, favorable psychometric properties of the PMHQ have been found for health sector workers in Mexico [55], higher education students in Portugal [7] and Colombia [56], preprofessional psychology practitioners in Peru [57], children between 9 and 12 years of age from Mexico [58], nursing students from Catalonia, Spain [59], nursing university professors in Spain [60], Colombian youth between 13 and 25 years [61] and Colombian youth aged 12 and over from Arequipa, Peru [62].

However, the psychometric analyses of the PMHQ in the various populations have shown diverse psychometric structures, which is attributed to the diversity of procedures used. In the factor analysis (FA), for example, in the validations with workers from the health sector in Mexico, with university students from Portugal, with Mexican children and with young people between the ages of 13 and 25 in Colombia, the internal structure was evaluated using the principal components method [7,55,58,61], which is a nonrecommended factor extraction procedure [63] that ignores measurement error and tends to inflate factor loading and explained variance and overestimates dimensionality [64]. In the validation processes of the PMHQ, the type of correlation calculated between items prior to the FA was not reported when the polychoric correlations were the most appropriate given that the responses to the items were ordinal, which makes it improbable that the requirement of normal distributions was met [64]. The factor estimation method was also omitted in the reviewed studies, with the unweighted least squares (ULS) method being the most appropriate because it is the most robust and recommended in the event of a possible violation of the assumption of normality [63,64]. The differentiated factorial structure in the reports can also be the result of the rotation method used; in this regard, Aparicio et al. [55], Aguilar [57] and Gómez-Acosta et al. [61] chose an orthogonal rotation method, which assumes independence between factors without allowing us to detect the correlation between them. Finally, only a few studies [57,58,59,62] performed a confirmatory factor analysis (CFA) to evaluate the structure found in the EFA. In relation to this, importantly, the purpose of the CFA is to evaluate hypothetical structures of the latent constructs resulting from the EFA and/or develop a better understanding of said structures; this procedure is a specific form of structural equation [54], in which researchers present a prespecified factor solution, which is evaluated in terms of how well it reproduces the sample covariance matrix of the measured variables [65].

On the other hand, in most of the reported studies, internal consistency was obtained through the alpha coefficient, except the validation carried out in Peru with participants older than 12 years, in which internal consistency was obtained through the omega coefficient. In this regard, it has been shown that the alpha coefficient has limitations, being susceptible to the number of items on the scale, the number of response options and the proportion of variance of the test [66].

### 1.5. Research Questions

The studies that have reported high correlations between PMHQ factors (e.g., [7,59,60,67]) have used exploratory factor analysis, which is a data-driven approach that lacks indices that allow evaluation of the goodness-of-fit of the models [68]. Confirmatory factor analysis, a theory-driven approach, can inflate the factor correlations and produce biased goodness-of-fit indices due to ignoring cross-loading [69]. When the correlation between factors ranges between 0.80 and 0.85, there is a risk of overfactoring the structure, which compromises its discriminant validity [70]. These facts motivated the first research question:Will the original structure of the PMHQ, based on previous studies (e.g., [7,59,60,67]) that reported high correlations between its factors, show poor fit?

Derived from the above, it is possible that a theory-and-data-driven modification of the PMHQ, based on the conceptual analysis of correlated factors, will allow the identification of a model with a better fit than the original structure. This motivated the second research question:Can a PMHQ structure with fewer than 6 factors (original structure) but more than 1 demonstrate an acceptable fit?

Finally, Cabarcas and Mendoza [56], through an exploratory factor analysis, detected that in the original 6-factor structure of the PMHQ, the first factor explained 67.18% of the variance. The remaining factors explained a value of less than 5%. This led them to evaluate a 1-factor structure for this instrument. This motivated the third and last research question:Will a unifactorial structure of the PMHQ yield an acceptable fit?

In this context, the primary objective of this study was to evaluate the psychometric properties of the PMHQ from a structural equations approach, that is, by estimating its structure through CFA. A secondary objective was to evaluate the structure of this instrument with 1 factor, or a smaller number of factors compared to the original structure but greater than 1.

## 2. Materials and Methods

### 2.1. Participants

Using a nonprobabilistic convenience sampling technique, a sample of 360 healthcare workers between the ages of 23 and 77 (mean (M) = 37.06; standard deviation (SD) = 10.79) with years of work experience between 0 and 53 years (M = 9.66) was recruited. The sample size is greater than the recommended minimum sample size for studies that use structural equation models, considering an anticipated effect size = 0.5, a statistical power level = 0.8 and a probability level = 0.05 for a model with 6 latent variables and 39 observed variable switches equal to n = 288 [71]. The inclusion criteria were workers in the medical or nursing area at the Hospital Infantil de México Federico Gómez and agreement to voluntarily participate in this study. The only exclusion criteria were not responding to three or more items of the PMHQ or closing the online form with the instruments, which caused the responses not to be recorded. Other sociodemographic and employment data are shown in Table 1. No type of material, economic or social incentive was offered to the participants for their collaboration in this study.

### 2.2. Measurement Instruments

#### 2.2.1. Sociodemographic Variables Questionnaire (Q-SV)

The sociodemographic variables questionnaire (Q-SV) was specifically designed for this study and collected sociodemographic (i.e., age, sex, marital status, educational level), labor (i.e., area and work experience, shift, second job in the public or private sector) and psychological variables (diagnosis of psychopathological condition and use of psychotropic drugs in the last 12 months) [72].

#### 2.2.2. Positive Mental Health Questionnaire

The PMHQ was originally designed and validated by Lluch-Canut [24] in 1999 and later validated with a Mexican population of health sector workers by Aparicio et al. [55]. It is made up of six factors with a total of 39 items: personal satisfaction (PS; 8 items); prosocial attitude (PA; 5 items); self-control (SC; 5 items); autonomy (AU; 5 items); problem-solving and self-actualization (PSSA; 9 items); and interpersonal relationship skills (IR; 7 items). The instrument is answered on a 4-point Likert-type scale. For the positive items, the scale is always/almost always (4), frequently (3), sometimes (2) and never/almost never (1). For negative items, the scale is reversed. The positive items are 4, 5, 11, 15–18, 20–23, 25–29, 32 and 35–37; the negative items are 1–3, 6–10, 12–14, 19, 24, 30, 31, 33, 34, 38 and 39. In the validation carried out by Aparicio et al. [55], the percentage of variance explained was 43.4%, and Cronbach’s α coefficient was 0.86 for the global instrument. These authors did not report the internal consistency obtained by factor. With the present population and prior to the psychometric analysis developed here, McDonald’s ω coefficient per-factor were PS = 0.82; PA = 0.58; SC = 0.84; AU = 0.83; PSSA = 0.82; IR = 0.73).

### 2.3. Design

This was a cross-sectional study of instrument types [73]. The construct validity of the PMHQ was assessed using confirmatory factor analysis, while its internal consistency was estimated using Cronbach’s alpha and the ordinal alpha.

### 2.4. Procedure

A questionnaire in printed format to be completed on paper with pencil or presented in an online format was answered by the participants between September 2022 and January 2023 in a single session of approximately 10 min. In both cases, the questionnaire included a cover letter explaining the objectives of this study and its benefits and requesting the voluntary participation of the target population. Those who agreed to participate in this study then completed the identification form and the PMHQ. Data collection using the printed questionnaire was carried out by a collaborator of the research group who was specially trained for this. The collaborator visited the medical area of the Hospital Infantil de México Federico Gómez and verbally requested the participation of the personnel present. Those who agreed to collaborate received the printed questionnaire, read and signed the cover letter, and then completed the instruments. The online form was used to collect data from the nursing area. A researcher was in charge of disseminating, among the personnel in this area, a link and/or a QR code that led directly to the online form hosted on Google Forms^®^. At the end of the cover letter, staff were asked to check a box labeled “I agree to participate in the study” or “I do not agree to participate in the study”. The choice of the first option led to the instruments, and the choice of the second allowed the closure of the online form without recording any data. The use of both formats was due to practical reasons for collecting the information, as well as evidence indicating the equivalence of the data collected in instruments completed on paper and pencil and online [74,75,76].

### 2.5. Ethical Considerations

This study is part of the research project “Salud mental positiva y su relación con la satisfacción y entusiasmo laboral y síndrome de quemarse por el trabajo durante la pandemia por COVID-19 en profesionales de la salud: un modelo predictivo” approved by the Research, Research Ethics and Biosafety Committee of the Hospital Infantil de México Federico Gómez (registration HIM-2021-054-FF). Likewise, it must be considered research with minimal risk for the participants in accordance with the Reglamento de la Ley General de Salud en Materia de Investigación para la Salud (Art. 3 Fracc. I, Art. 4, Art. 6, Título II Cap. I, Art. 17 Fracc. II) and its update published in the Diario Oficial de la Federación (2 April 2014) and based on the Norma Oficial Mexicana NOM-012-SSA3-2012 (Section 5 numerals 5.3 to 5.13 and 5.15), which establishes the criteria for the execution of research projects for health in human beings. This study was performed in adherence to the Declaration of Helsinki updated in 2013, to current ethical considerations in Mexico for research with humans [77] and to those outlined by the American Psychological Association [78]. Participation in this study was voluntary and did not involve any kind of incentive.

### 2.6. Data Analytic Strategy

The data from the PMHQ were analyzed using the R v.4.3.1 program [79] and its RStudio v.2023.06.1 interface [80]. The packages dplyr v.1.1.2 [81], mice v.3.16.0 [82], naniar v.1.0.0 [83], psych v.2.3.9 [84], MVN v.5.9 [85], stats v.4.3.1 [79], PerformanceAnalytics v.2.0.4 [86], lavaan v.0.6-16 [87], semPlot v.1.1.6 [88] and misty v.0.5.3 [89] were used.

#### 2.6.1. Data Processing

The presence of missing data was evaluated. Little’s [90] missing completely at random (MCAR) test was used to detect their structure. In the case of missing data that were not missing completely at random, a listwise deletion was employed. This was performed to avoid incorporating bias into the data analysis. However, the unobserved data that could have caused said missing data were analyzed. With a database free of missing data, the presence of multivariate outliers was analyzed using the Mahalanobis distance (D2; [91]) with a cutoff point established by the χ^2^ value with degrees of freedom equal to the number of items in the PMHQ (i.e., [25]) and a *p* < 0.001, although these were not withdrawn unless they revealed consistent patterns of mechanical response (c.f., [92]).

#### 2.6.2. Descriptive Statistics of the Items

The mean and standard deviation of each item of the PMHQ and its coefficient of asymmetry and kurtosis were estimated. Univariate and multivariate normality was analyzed using the Anderson–Darling test [93] and Mardia’s skewness and kurtosis [94], respectively. The report of multivariate outliers and multivariate normality supplement the scarce information about this variable in structural equation model studies [95]. The polychoric correlation of all 39 items of the PMHQ was estimated with the software FACTOR v.12.04.04 [96].

#### 2.6.3. Confirmatory Factor Analysis

The construct validity of the PMHQ was assessed using CFA with the weighted least squares means and variance adjusted (WLSMV) estimator with data from the polychoric correlation and asymptotic covariance matrix [97,98]. Items with *p* < 0.05 were retained, and whether they met the criterion of mean factor loading ≥0.70 per factor was verified [99]. The correlations between factor scores generated from CFA were reported. The fit of the model was evaluated using the χ^2^ test, χ^2^/df, root mean square error of approximation (RMSEA), weighted root mean square residual (WRMR), comparative fit index (CFI) and Tucker–Lewis index (TLI). The following fit values were considered acceptable: χ^2^/df ≤ 5, RMSEA ≤ 0.08 (90% CI < 0.10), WRMR ≤ 1, CFI ≥ 0.90, TLI ≥ 0.90; and excellent fit: χ^2^/df ≤ 2, RMSEA ≤ 0.05, CFI ≥ 0.95, TLI ≥ 0.95 [100]. Goodness-of-fit indices were evaluated sequentially, and in the cases of values below what was acceptable, appropriate respecifications were made to the model. For each respecification, statistical criteria (modification indices and factorial saturation of each item) and theoretical criteria (item-factor and factor-construct conceptual coherence) were considered to maintain the conceptual value of the instrument.

#### 2.6.4. Internal Consistency

Internal consistency was evaluated using Cronbach’s alpha (α) to compare studies, and McDonald’s ω for estimating reliability more accurately than α [101]. For Cronbach’s α coefficient, a value ≥ 0.70 was considered acceptable [102], and for McDonald’s ω, a value ≥ 0.80 [103] and ≤0.94 [104] were considered acceptable.

## 3. Results

### 3.1. Data Processing

In a database with 360 observations for 39 variables, 34 missing data points were found, corresponding to 0.2% of the total (see Appendix A). There were no missing data completely at random, according to Little’s test (χ^2^ = 782, *p* = 0.03); therefore, each observation with at least one missing data point was removed from the dataset. This eliminated 24 observations coming from the collection of information made with the printed questionnaire and left n = 336 for posterior analysis. After this, 22 multivariate outliers were identified. However, visual inspection of the cases did not reveal the presence of systematic patterns of mechanical response, so the observations were kept for further analysis.

### 3.2. Descriptive Analysis, Univariate and Multivariate Normality and Polychoric Correlation

Table 2 presents the descriptive analyses of the items of the PMHQ. There was no evidence of univariate (Anderson–Darling test, minimum value = 22.19 and maximum value = 101.29, *p* < 0.01) or multivariate normality (Mardia skewness = 24,045.96; Mardia kurtosis = 53.83, *p* < 0.01). In the Appendix A, the polychoric correlation matrix of all 39 items is shown (Appendix A).

### 3.3. Confirmatory Factor Analysis

#### 3.3.1. Original Structure with 6 Factors

Confirmatory factor analysis of the original version with 6 factors of the PMHQ did not show a positive definite covariance matrix. Table 3 (without constraints) shows the correlation matrix between the six factors and reveals a high correlation for Factor 3 with Factor 5 and for Factor 2 and Factor 6. Additionally, an eigenvalue was negative (i.e., −0.0007). A covariation constraint for both pairs of factors was fixed at 0.05, and the model was estimated again. The covariance matrix was now positive definite with all eigenvalues > 0. The correlation matrix between factors is shown in Table 3 (with constraints). All factor loadings were significant (*p* < 0.001). The standardized λ ranged between 0.12 and 0.84 (see Appendix A), and all covariances between factors were significant (range for standardized values = 0.16 to 0.79, *p* < 0.05). The goodness-of-fit indices and mean factor loading for this structure are shown in Table 4. Except for the χ^2^ test and WRMR, the rest were acceptable. All Mλ were <0.70.

#### 3.3.2. Structure with 4 Factors

In this model, Factor 2 (prosocial attitude) and Factor 6 (interpersonal relationship skills), the new Factor 2, as well as Factor 3 (self-control) and Factor 5 (problem-solving and self-actualization) and the new Factor 3 were integrated into one. The correlation matrix between factors is shown in Table 5. All factor loadings were significant (*p* < 0.001). The standardized λ ranged between 0.33 and 0.84 (see Appendix A), and all covariances between factors were significant (range for standardized values = 0.39 to 0.65, *p* < 0.001). The goodness-of-fit indices and mean factor loading for this structure are shown in Table 4. Except for χ^2^/df, the rest were unacceptable. The Mλ was <0.70.

#### 3.3.3. Structure with 1 Factor

All factor loadings were significant (*p* < 0.001). The standardized λ ranged between 0.23 and 0.76 (see Appendix A). The goodness-of-fit indices and mean factor loading for this structure are shown in Table 4. Except for the χ^2^ test and WRMR, the rest were acceptable. All Mλ were <0.70.

### 3.4. Internal Consistency

Table 6 shows the values for Cronbach’s α and McDonald’s ω for the structures from the PMHQ with six and four factors and one factor. Based only on McDonald’s ω, for the 6-factor structure, the internal consistency was acceptable only for Factor 1 and Factor 3 to Factor 5. For the 4-factor structure, all factors but Factor 2 were acceptable. The one-factor structure was also acceptable.

## 4. Discussion

The purpose of the present study was to assess the psychometric properties of the PMHQ from three structures using a structural equations approach. For objective 1, the original structure was evaluated with 6 factors. The results showed acceptable fit and acceptable internal consistency for 4 of its 6 factors. For objective 2, a structure with fewer than 6 factors but with more than 1 was evaluated. The results indicated that a structure with 4 factors presented an acceptable fit and acceptable internal consistency for 3 of its 4 factors. For objective 3, a structure of only 1 factor was evaluated, which showed an unacceptable fit but acceptable internal consistency.

### 4.1. Factor Structure

#### 4.1.1. Six-Factor Positive Mental Health Questionnaire

The original structure with 6 factors for the PMHQ has been reported in research using exploratory [24] as well as confirmatory factor analyses [59]. In this study, a confirmatory factor analysis detected a correlation of 0.88 between the factors of prosocial attitude (Factor 2) and interpersonal relationship skills (Factor 6) and 0.83 between the factors of self-control (Factor 3) and problem-solving and self-actualization (Factor 5). Because the correlation matrix was not positive definite, there was a risk of obtaining biased goodness-of-fit indices. After constraining the covariation between these factors, the correlation was less than 0.80, and the correlation matrix was positive definite. Then, the fit of the model was acceptable, and its goodness-of-fit indices were comparable to those obtained by Roldán-Merino et al. [59]. The factor loadings were significant, although they presented a wide variability and a mean of less than 0.70 in each factor. The above may suggest the presence of weak factors in the PMHQ, which would imply a diminished influence on the observable variables it integrates [105]. Future studies must investigate this fact.

#### 4.1.2. Four-Factor Positive Mental Health Questionnaire

The correlation between factors 2 and 6 and factors 3 and 5 is greater than 0.80. According to Watkins [70], this can cause overfactoring in the structure of a measurement instrument. Under this premise, the review of the items of factors 3 “self-control” and 5 “problem-solving and self-actualization” and 2 “prosocial attitude” and 6 “interpersonal relationship skills” indicated a conceptual coherence that allowed the integration of each pair into a single factor. This involved transforming the PMHQ from a 6- to 4-factor scale. In favor of this conceptual integration, there are studies that have found an association between self-control and problem-solving [106] and between prosocial behaviors and interpersonal relationships [107]. The model with 4 factors of the PMHQ was composed of Factor 1 “personal satisfaction” (8 items), Factor 2 “prosocial attitude and interpersonal relationship skills” (8 items), Factor 3 “self-control, problem-solving and self-actualization” (14 items) and Factor 4 “autonomy” (5 items). This structure with 4 factors presented acceptable goodness-of-fit indices. The factor loads in this model present less variability with respect to the 6-factor model, but similar to this one, the mean factor loading did not reach the expected value of 0.70.

These data are comparable to those of the confirmatory factor analysis evaluation of the PMHQ developed by Roldán-Merino et al. [59] and present a more parsimonious model. This implies that the construct evaluated by this version of the PMHQ is similar to that of the original version of the instrument.

#### 4.1.3. One-Factor Positive Mental Health Questionnaire

When all of the factors were removed, the model presented an unacceptable fit. The results obtained suggest that this 1-factor structure of the PMHQ is not adequate to assess positive mental health. This statement is based on the absence of an acceptable fit in its indices, the variability in the factor loads and the fact that the mean factor load of the single factor did not reach the criterion value of 0.70.

### 4.2. Internal Consistency of the 1-Factor, 4-Factor and 6-Factor Positive Mental Health Questionnaire

For the structure with 6 factors, from the calculation of McDonald’s ω, an acceptable internal consistency was found in 4 of them. In the case of the model with 4 factors, there was an acceptable internal consistency in 3 of them, which indicates that the results obtained from these factors are repeatable and consistent. Factors with unacceptable internal consistency suggest that the items are heterogeneous depending on what they claim to measure, since the omega is based on a factor analysis model that examines the interrelationships between elements and subsets of elements [108]. Factor 2 increases its internal consistency in the model with 4 factors, which supports the idea of uniting factors 6 and 2 and demonstrates the relationships among emotional competence, interpersonal relationships and prosocial behaviors, as detected in a study by Pung et al. [109]. This study also pointed out that emotional competence and interpersonal relationships with peers promotes psychosocial behavior.

With respect to the Cronbach’s α obtained in previous studies in the case of the model with 6 factors, a systematically lower value in factor 2 is notable [14,17,18,19]. This again justifies the union of the factor related to interpersonal relationships and prosocial behaviors in which the first acts as a mediator of the second; it is also based on the relationship between prosocial behaviors and variables existing in positive interpersonal behaviors such as empathy, desire for adventure, positive self-concept and self-esteem, and self-perceived health [110].

### 4.3. Limitations and Strengths of this Study

This study has several limitations. The first refers to the sample. The sampling technique used was nonrandom for convenience, which limits the generalizability of the results. Second, only personnel from the medical and nursing areas were included, although they are the ones who mainly have contact with users of healthcare services. The most important limitation is the omission of other procedures, such as concurrent or divergent validity, in the validation of the PMHQ. This was because the final goal of this study was to detect the best possible structure for this scale and, from there, conduct new studies that complete its validation process. Despite its limitations, this study also has several strengths. Among the methodological strengths is the use of a rigorous statistical procedure that used robust methods and that satisfied statistical assumptions for its use in the type of data analyzed [111]. Likewise, the variations analyzed in the structure of the PMHQ were made based on conceptual aspects and guided by statistical data that maintain the conceptual coherence of this instrument [51].

## 5. Conclusions

Our findings showed that the 6-factor and 4-factor structures of the PMHQ presented acceptable adjustments while keeping intact the concept developed by Lluch-Canut [24] for this scale. However, the factor loadings are less variable for the 4-factor structure, which could suggest a better representation of the concept evaluated by each factor. Additionally, this last structure presents an acceptable internal consistency in three of its four factors. In conclusion, both the 6- and 4-factor structures of the PMHQ are valid instruments for evaluating positive mental health in healthcare personnel in Mexico.

## Figures and Tables

**Table 1 healthcare-11-03041-t001:** Participant data for the sociodemographic variable questionnaire.

Variable	Category	n (%)
Sex	Female	273 (75.8)
	Male	86 (23.9)
	Missing	1 (0.3)
Marital Status	Single	217 (60.3)
	Married	86 (23.9)
	Consensual union	40 (11.1)
	Divorced	9 (2.5)
	Separated	3 (0.8)
	Widowed	5 (1.4)
Educational level	Technical school	14 (3.9)
	Bachelor’s degree	121 (33.6)
	Specialty	110 (30.6)
	Subspecialty	89 (24.7)
	Master’s degree	22 (6.1)
	PhD	3 (0.8)
	Missing	1 (0.3)
Study Area	Medical	237 (65.8)
	Nursing	123 (34.2)
Shift	Morning	178 (49.4)
	Evening	25 (6.9)
	Night	30 (8.3)
	Other	127 (35.3)
Second job in public sector	Yes	31 (8.6)
	No	329 (91.4)
Second job in private practice	Yes	49 (13.6)
	No	311 (86.4)
Diagnosis of psychopathology in the past 12 months	Yes	62 (17.2)
	No	298 (82.8)
Use of psychotropic drugs in the past 12 months	Yes	68 (18.9)
	No	292 (81.1)

**Table 2 healthcare-11-03041-t002:** Descriptive statistics for the items of the Positive Mental Health Questionnaire.

Item No.	Item Content	M	SD	S	K
	Factor 1: Personal satisfaction				
4	I like myself as I am.	3.51	0.72	−1.40	1.40
6	I feel like I am about to explode.	3.30	0.72	−1.00	1.21
7	I find life to be boring and monotonous.	3.61	0.68	−1.94	3.75
12	I view the future with pessimism.	3.76	0.57	−2.70	7.94
14	I see myself as less important than those around me.	3.76	0.59	−2.85	8.23
31	I feel inept and useless.	3.85	0.51	−4.01	16.89
38	I feel unsatisfied with myself.	3.40	0.98	−1.52	1.01
39	I feel unsatisfied with the way I look.	3.38	0.82	−1.29	1.05
	Factor 2: Prosocial attitude				
1	I find it especially difficult to accept others when their attitudes are different from mine.	3.40	0.67	−1.02	1.31
3	I find it particularly difficult to listen to people tell me their problems.	3.65	0.64	−2.16	5.25
23	I feel that I am someone to be trusted.	3.86	0.45	−3.79	16.59
25	I consider the needs of others.	3.23	0.84	−0.71	−0.54
37	I like to help others.	3.65	0.62	−1.87	3.43
	Factor 3: Self-control				
2	Problems often cause me to feel blocked.	3.44	0.66	−1.20	2.01
5	I am able to control myself when I feel negative emotions.	3.12	0.82	−0.52	−0.57
21	I am able to control myself when I have negative thoughts.	3.28	0.80	−0.76	−0.41
22	I am able to maintain a high level of self-control in conflictive situations in my life.	3.33	0.71	−0.62	−0.66
26	When I experience unpleasant external pressure, I am able to maintain my personal balance.	3.24	0.77	−0.79	0.14
	Factor 4: Autonomy				
10	I worry a lot about what others think of me.	3.32	0.77	−1.04	0.75
13	The opinions of others have a strong influence on me when I have to make decisions.	3.49	0.67	−1.24	1.46
19	It troubles me when people criticize me.	3.31	0.79	−0.96	0.28
33	I find it hard to hold my own opinions.	3.57	0.71	−1.67	2.29
34	When I have to make big decisions, I feel very unsure of myself.	3.34	0.79	−1.25	1.31
	Factor 5: Problem-solving and self-actualization				
15	I am able to make decisions on my own.	3.68	0.69	−2.35	5.13
16	I try to look for the positive side when bad things happen to me.	3.45	0.69	−0.96	0.10
17	I try to improve myself as a person.	3.73	0.59	−2.27	4.92
27	When there are changes in my surroundings, I try to adapt to them.	3.60	0.63	−1.43	1.49
28	In the face of a problem, I am able to ask for information.	3.59	0.66	−1.52	1.71
29	I find changes in my daily routine to be stimulating.	3.23	0.76	−0.52	−0.71
32	I try to develop my abilities to the maximum.	3.61	0.64	−1.73	3.03
35	I am able to say no when I want to.	3.15	0.92	−0.69	−0.67
36	When I am faced with a problem, I try to find possible solutions.	3.73	0.54	−2.22	5.62
	Factor 6: Interpersonal relationship skills				
8	I find it particularly difficult to provide emotional support to others.	3.48	0.71	−1.39	1.85
9	I find it hard to establish deep and satisfying interpersonal relationships with some people.	3.50	0.71	−1.35	1.42
11	I feel that I have a strong ability to put myself in the shoes of others and to understand their responses.	3.04	0.93	−0.57	−0.73
18	I consider myself to be a good professional.	3.74	0.54	−2.34	6.25
20	I think that I am a sociable person.	3.19	0.84	−0.67	−0.52
24	I find it particularly hard to understand the feelings of others.	3.40	0.73	−1.10	0.88
30	I find it hard to relate openly with my teachers/bosses.	3.35	0.83	−1.22	0.84

Notes: K: kurtosis; M: mean; S: skewness; SD: standard deviation.

**Table 3 healthcare-11-03041-t003:** Correlation between factors of the Positive Mental Health Questionnaire with 6 factors.

	Without Constraint					With Constraint				
Factors	F1	F2	F3	F4	F5	F1	F2	F3	F4	F5
F2	0.34					0.31				
F3	0.60	0.53				0.60	0.49			
F4	0.65	0.22	0.61			0.65	0.16	0.61		
F5	0.53	0.79	0.83	0.52		0.53	0.79	0.79	0.50	
F6	0.61	0.88	0.59	0.47	0.67	0.61	0.73	0.59	0.47	0.67

Notes: F: Factor.

**Table 4 healthcare-11-03041-t004:** Goodness-of-fit indices and mean factor loading for the structures evaluated from the Positive Mental Health Questionnaire.

Factors	χ^2^	χ^2^/df	RMSEA (90% CI)	WRMR	CFI	TLI	Mλ					
							F1	F2	F3	F4	F5	F6
F6	2346.27 **	3.40	0.085 (0.081 to 0.089)	1.65	0.913	0.906	0.67	0.52	0.72	0.68	0.61	0.52
F4	2018.81 **	2.90	0.075 (0.072 to 0.079)	1.53	0.931	0.926	0.67	0.52	0.68	0.65		
F1	2898.64 **	4.12	0.097 (0.093 to 0.100)	1.86	0.885	0.878	0.52					

Notes: F: Factor. Note: ** *p* < 0.05.

**Table 5 healthcare-11-03041-t005:** Correlation between factors of the Positive Mental Health Questionnaire with 4 factors.

	F1	F2	F3
F2	0.53		
F3	0.65	0.39	
F4	0.59	0.70	0.58

Notes: F: Factor.

**Table 6 healthcare-11-03041-t006:** Internal consistency of the structures evaluated in the Positive Mental Health Questionnaire.

Factors	F1	F2	F3	F4	F5	F6
6	α = 0.75; ω = 0.82	α = 0.47; ω = 0.58	α = 0.78; ω = 0.84	α = 0.69; ω = 0.83	α = 0.77; ω = 0.82	α = 0.59; ω = 0.73
4	α = 0.75; ω = 0.82	α = 0.70; ω = 0.75	α = 0.85; ω = 0.87	α = 0.69; ω = 0.83		
1	α = 0.89; ω = 0.90					

Notes: α: Cronbach’s alpha; ω: McDonald’s omega.

## Data Availability

The raw data supporting the conclusions of this article will be made available by the authors without undue reservation.

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
