# Peer review of "Positive Mental Health Questionnaire (PMHQ) for Healthcare Workers: A Psychometric Evaluation"

_healthcare, 2023, doi:10.3390/healthcare11233041_

Round 1
Reviewer 1 Report
Comments and Suggestions for Authors
The submitted manuscript analyzes the positive mental health questionnaire (PMHQ) administered to Mexican health workers. The purpose of the paper is clear. I want to comment on several aspects that should be considered in a manuscript revision.
Major comments:
1. According to the MDPI guidelines, line numbers should be included. This would ease the review.
2. Sect. 2.2.: It is unclear what is meant by “without data on its consistency by factor.”? Please clarify.
3. Sect. 2.6.: Provide references for R, Rstudio, and all used R packages.
4. Sect. 2.6.1.: The title “Data Engineering” sounds strange. “Data processing” would better fit, in my view.
5. Sect. 2.6.1.: I doubt the different missing data mechanisms can be explored or detected “through their visual representation.” Missing at random (MAR) can only be assumed, while there exist statistical tests for missing completely at random (MCAR). Moreover, imputing missing values by its mode is one of the worst missing data treatments. Use the mice package with predictive mean matching as an appropriate imputation method. Given the fact that missing data proportions are low, single imputation would suffice in your case.
6. Sect. 2.6.2.: I do not know why it is useful to assess multivariate normality. If you analyze the data (i.e., the items) with polychoric correlations, then you do not rely on the normality assumption.
7. Sect. 2.6.3.: You state that you report tau instead of Pearson correlations between the factor variables. The reasoning is strange. Nonnormally distributed items do not directly translate into nonnormally distributed factor variables. Moreover, I thought that you applied a confirmatory factor analysis (CFA) to your items based on polychoric correlation. Then, in Sect. 3.3., you can directly report the correlations between the factors from the CFA output. Using factor scores in a two-step will produce biased correlations unless you make appropriate adjustments (but I doubt that you did so). Correlations between latent variables make sense even if they are nonnormally distributed.
8. Sect. 2.6.2.: Exclude anything related to multivariate outliers. It does not make sense to remove observations just due to statistical reasons. Even the employed statistical models do not require outlier-free observations.
9. Sect. 2.6.4.: I do not see why items with a loading smaller than .40 should be removed. More critically, I think that sampling errors are too large in your relatively small sample to make reliable statistical inferences about thresholds on factor loadings. Compute a standard error for the respective loadings and ensure that the confidence interval does not cover the desired cutoff of .40.
10. Table 2: The first column should be labeled “Item.” What do the suffixes “I” and “N” in this column mean?
11. Table 2: Left align the second column. I think that “Item content” or “Item formulation” would be a more adequate column name.
12. Table 2: Also report the observed frequencies of the four item categories in the table. In addition, include the polychoric correlation matrix of all 39 items in a supplementary material or make your dataset publicly available.
13. Sect. 3.4.: You repeatedly argue that you cannot rely on the six-factor solution because the covariance matrix is “positive defined.” First, the correct term is “positive definite”. Second, I suppose the warning is related to the covariance matrix of parameter estimates (i.e., related standard errors). Issues can be circumvented by using constrained estimation. It could also be the case that you have sparse data (i.e., you have no or only a few observations in some categories for some items). These data issues must not dictate the choice of your final model. That is, stick to your six-factor model and investigate the cause of the reported issues. Moreover, you reason that some factors are “highly correlated” and should be merged. However, the reported factor correlations do not exceed typical cutoffs for high correlations (e.g., .85). Hence, using fewer factors is unjustified in your case.
14. Using ordinal alpha for ordinal items is an inappropriate reliability measure. Please consult Chalmers (2018), Green and Yang (2009), and https://github.com/simsem/semTools/issues/48. Remove anything related to ordinal alpha in the manuscript. Use a proper reliability statistic for ordinal data (see the discussion in Github in the link above).
Minor comments:
15. Abstract, Sect. 2.1.: Write M=37.06 and SD=10.79 (i.e., do not write mathematical symbols in parenthesis).
16. Abstract: Write “one-factor” instead of “unifactorial” structure.
17. Sect. 2.2.: Is it better to say that Likert-scale items with “4 points” instead of “4 levels” are used?
18. Sect. 2.6.4.: Do not write lambda in parenthesis.
19. Sect. 3.1.: Remove the information about “data points.”
20. Sect. 3.2.: Write “p < .01”.
References:
Chalmers, R. P. (2018). On misconceptions and the limited usefulness of ordinal alpha. Educational and Psychological Measurement, 78(6), 1056-1071.
Green, S.B., Yang, Y. Reliability of Summed Item Scores Using Structural Equation Modeling: An Alternative to Coefficient Alpha. Psychometrika 74, 155–167 (2009). https://doi.org/10.1007/s11336-008-9099-3
---
Author Response
Comments and Suggestions for Authors
The submitted manuscript analyzes the positive mental health questionnaire (PMHQ) administered to Mexican health workers. The purpose of the paper is clear. I want to comment on several aspects that should be considered in a manuscript revision.
Major comments:
According to the MDPI guidelines, line numbers should be included. This would ease the review.
Response:
Line numbers have been included.
Major comments:
Sect. 2.2.: It is unclear what is meant by “without data on its consistency by factor.”? Please clarify.
Response:
These authors did not report the internal consistency obtained by factor.
Change:
2.2.2. Positive Mental Health Questionnaire
The PMHQ was originally designed and validated by Lluch-Canut [24] in 1999 and later validated with a Mexican population of health sector workers by Aparicio et al. [41]. It is made up of six factors with a total of 39 items: personal satisfaction (PS; 8 items); prosocial attitude (PA; 5 items); self-control (SC; 5 items); autonomy (AU; 5 items); problem-solving and self-actualization (PSSA; 9 items); and interpersonal relationship skills (IR; 7 items). The instrument is answered on a 4-point Likert-type scale. For the positive items, the scale is always/almost always (4), frequently (3), sometimes (2) and never/almost never (1). For negative items, this scale is reversed. The positive items are 4, 5, 11, 15-18, 20-23, 25-29, 32, and 35-37; the negative items are 1-3, 6-10, 12-14, 19, 24, 30, 31, 33, 34, 38, and 39. In the validation carried out by Aparicio et al. [41], the percentage of variance explained was 43.4%, and Cronbach's α coefficient was 0.86 for the global instrument. These authors did not report the internal consistency obtained by factor. With the present population and prior to the psychometric analysis developed here, McDonald's omega (ω) coefficient per factor was PS = 0.82; PA = 0.58; SC = 0.84; AU = 0.83; PSSA = 0.82; IR = 0.73).
Major comments:
Sect. 2.6.: Provide references for R, Rstudio, and all used R packages.
Response:
References to R, RStudio and the libraries used have been included.
Change:
The data from the PMHQ were analyzed using the R v.4.3.1 program (R Core Team, 2023) and its RStudio v.2023.06.1 interface (RStudio Team, 2020). The packages dplyr (Wickham et al., 2023), mice (van Buuren & Groothuis-Oudshoorn, 2011), naniar (Tierney & Cook, 2023), psych (Revelle, 2023), MVN (Korkmaz et al., 2014), stats (R Core Team, 2023), PerformanceAnalytics (Peterson & Carl, 2020), lavaan (Rosseel, 2012), semPlot (Epskamp, 2022) and misty (Yanagida, 2023) were used.
Major comments:
Sect. 2.6.1.: The title “Data Engineering” sounds strange. “Data processing” would better fit, in my view.
Response:
The title “Data Engineering” was replaced by “Data processing”.
Change:
2.6.1. Data processing
The presence of missing data was evaluated, and Little's missing completely at random (MCAR) test (1988) was used to detect their structure. In the case of missing data that were not missing completely at random, a listwise deletion was employed. With a database free of missing data, the presence of multivariate outliers was analyzed using the Mahalanobis distance (D2; [60]) with a cutoff point established by the χ2 value with degrees of freedom equal to the number of items in the PMHQ (i.e., [40]) and a p < 0.001, although these were not withdrawn unless they revealed consistent patterns of mechanical response (c.f., [61]).
Major comments:
I doubt the different missing data mechanisms can be explored or detected “through their visual representation.” Missing at random (MAR) can only be assumed, while there exist statistical tests for missing completely at random (MCAR). Moreover, imputing missing values by its mode is one of the worst missing data treatments. Use the mice package with predictive mean matching as an appropriate imputation method. Given the fact that missing data proportions are low, single imputation would suffice in your case.
Response:
Little's missing completely at random (MCAR) test was used. With this test, it was discovered that the missing data werenot completely missing at random. Due to this, it was decided to eliminate each observation with at least one missing data point. Even so, the sample size remained above the minimum necessary, as justified in Section 2.1. Furthermore, this implied a new estimation for all of the analyses.
Change:
The presence of missing data was evaluated, and Little's missing completely at random (MCAR) test (1988) was used to detect their structure. In the case of missing data that were not missing completely at random, a listwise deletion was employed. With a database free of missing data, the presence of multivariate outliers was analyzed using the Mahalanobis distance (D2; [60]) with a cutoff point established by the χ2 value with degrees of freedom equal to the number of items in the PMHQ (i.e., [40]) and a p < 0.001, although these were not withdrawn unless they revealed consistent patterns of mechanical response (c.f., [61]).
Major comments:
I do not know why it is useful to assess multivariate normality. If you analyze the data (i.e., the items) with polychoric correlations, then you do not rely on the normality assumption.
Response:
I agree with the observation. Because the extraction method is robust and based on polychoric correlations, multivariate normality is dispensed with. However, recent studies have not reported these variables (i.e., multivariate normality and multivariate outliers) given the effect they can have on the structure of a structural equation model (e.g., Lai & Zhang, 2017). These authors contend that “Despite the documented impact of outliers and influential observations, detection and diagnosis of such observations were rarely performed and reported in real research, and the use of SEM methods that are robust to data contamination has been scarce” p. 3).
Change:
2.6.2. Descriptive Analysis, Univariate and Multivariate Normality and Polychoric Correlation
The mean and standard deviation of each item of the PMHQ and its coefficient of asymmetry and kurtosis were estimated. Univariate and multivariate normality was analyzed using the Anderson‒Darling test (Hawkins, 2023) and Mardia’s skewness and kurtosis (Wulandari et al., 2021), respectively. The report of multivariate outliers and multivariate normality supplement the scarce information about this variable in structural equation model studies (c.f., Lai & Zhang, 2017). The polychoric correlation of all 39 items of the PMHQ was estimated with the software FACTOR v.12.04.04 (Lorenzo-Seva & Ferrando, 2006; https://psico.fcep.urv.cat/utilitats/factor/index.html).
Major comments:
Sect. 2.6.3.: You state that you report tau instead of Pearson correlations between the factor variables. The reasoning is strange. Nonnormally distributed items do not directly translate into nonnormally distributed factor variables. Moreover, I thought that you applied a confirmatory factor analysis (CFA) to your items based on polychoric correlation. Then, in Sect. 3.3., you can directly report the correlations between the factors from the CFA output. Using factor scores in a two-step will produce biased correlations unless you make appropriate adjustments (but I doubt that you did so). Correlations between latent variables make sense even if they are nonnormally distributed.
Response:
The correlation between the latent factors was reported based on the data provided by the CFA.
Change:
2.6.3. Confirmatory Factor Analysis
The construct validity of the PMHQ was assessed using CFA with the weighted least squares means and variance adjusted (WLSMV) estimator with data from the polychoric correlation and asymptotic covariance matrix [62, 63]. Items with p < 0.05 were retained, and whether they met the criterion of mean factor loading ≥.70 per factor was verified [64]. The correlations between factor scores generated from CFA were reported. The fit of the model was evaluated using the χ2 test, χ2/df, root mean square error of approximation (RMSEA), weighted root mean square residual (WRMR), comparative fit index (CFI) and Tucker‒Lewis index (TLI). The following fit values were considered acceptable: χ2/df ≤ 5, RMSEA ≤ 0.08 (90% CI < 0.10), WRMR ≤ 1, CFI ≥ 0.90, TLI ≥ 0.90; and excellent fit: χ2/df ≤ 2, RMSEA ≤ 0.05, CFI ≥ 0.95, TLI ≥ 0.95 [65]. Goodness-of-fit indices were evaluated sequentially, and in the cases of values below what was acceptable, appropriate respecifications were made to the model. For each respecification, statistical criteria (modification indices and factorial saturation of each item) and theoretical criteria (item-factor and factor-construct conceptual coherence) were considered to maintain the conceptual value of the instrument.
Table 3. Correlation between factors of the Positive Mental Health Questionnaire with 6 factors
|
|
Without constraint |
With constraint |
||||||||
|
Factors |
F1 |
F2 |
F3 |
F4 |
F5 |
F1 |
F2 |
F3 |
F4 |
F5 |
|
F2 |
0.34 |
|
|
|
|
0.31 |
|
|
|
|
|
F3 |
0.60 |
0.53 |
|
|
|
0.60 |
0.49 |
|
|
|
|
F4 |
0.65 |
0.22 |
0.61 |
|
|
0.65 |
0.16 |
0.61 |
|
|
|
F5 |
0.53 |
0.79 |
0.83 |
0.52 |
|
0.53 |
0.79 |
0.79 |
0.50 |
|
|
F6 |
0.61 |
0.88 |
0.59 |
0.47 |
0.67 |
0.61 |
0.73 |
0.59 |
0.47 |
0.67 |
Abbreviations: F: Factor.
Table 5. Correlation between factors of the Positive Mental Health Questionnaire with 4 factors
|
|
F1 |
F2 |
F3 |
|
F2 |
0.53 |
|
|
|
F3 |
0.65 |
0.39 |
|
|
F4 |
0.59 |
0.70 |
0.58 |
Abbreviations: F: Factor.
Major comments:
Sect. 2.6.2: Exclude anything related to multivariate outliers. It does not make sense to remove observations just due to statistical reasons. Even the employed statistical models do not require outlier-free observations.
Response:
As mentioned before, it is a complaint in recent literature that these variables are not reported (see Lai & Zhang, 2017, p. 3). Therefore, despite the use of a robust extraction method, this information was maintained. In addition, Reviewer 2 requested it.
Change:
2.6.2. Descriptive Analysis, Univariate and Multivariate Normality and Polychoric Correlation
The mean and standard deviation of each item of the PMHQ and its coefficient of asymmetry and kurtosis were estimated. Univariate and multivariate normality was analyzed using the Anderson‒Darling test (Hawkins, 2023) and Mardia’s skewness and kurtosis (Wulandari et al., 2021), respectively. The report of multivariate outliers and multivariate normality supplements the scarce information about this variable in structural equation model studies (c.f., Lai & Zhang, 2017). The polychoric correlation of all 39 items of the PMHQ was estimated with the software FACTOR v.12.04.04 (Lorenzo-Seva & Ferrando, 2006; https://psico.fcep.urv.cat/utilitats/factor/index.html).
Major comments:
Sect. 2.6.4.: I do not see why items with a loading smaller than .40 should be removed. More critically, I think that sampling errors are too large in your relatively small sample to make reliable statistical inferences about thresholds on factor loadings. Compute a standard error for the respective loadings and ensure that the confidence interval does not cover the desired cutoff of .40.
Response:
The retention of the items was based on their level of significance (p < 0.05).
Change:
2.6.3. Confirmatory Factor Analysis
The construct validity of the PMHQ was assessed using CFA with the weighted least squares means and variance adjusted (WLSMV) estimator with data from the polychoric correlation and asymptotic covariance matrix [62, 63]. Items with p < 0.05 were retained, and whether they met the criterion of mean factor loading ≥ .70 per factor was verified [64]. The correlations between factor scores generated from CFA were reported. The fit of the model was evaluated using the χ2 test, χ2/df, root mean square error of approximation (RMSEA), weighted root mean square residual (WRMR), comparative fit index (CFI) and Tucker‒Lewis index (TLI). The following fit values were considered acceptable: χ2/df ≤ 5, RMSEA ≤ 0.08 (90% CI < 0.10), WRMR ≤ 1, CFI ≥ 0.90, TLI ≥ 0.90; and excellent fit: χ2/df ≤ 2, RMSEA ≤ 0.05, CFI ≥ 0.95, TLI ≥ 0.95 [65]. Goodness-of-fit indices were evaluated sequentially, and in the case of values below what was acceptable, appropriate respecifications were made to the model. For each respecification, statistical criteria (modification indices and factorial saturation of each item) and theoretical criteria (item-factor and factor-construct conceptual coherence) were considered to maintain the conceptual value of the instrument.
Major comments:
Table 2: The first column should be labeled “Item.” What do the suffixes “I” and “N” in this column mean?
Response:
The suffixes I and N were eliminated. Section 2.2.2 clarifies which items are positive and which are negative.
Change:
2.2.2. Positive Mental Health Questionnaire
The PMHQ was originally designed and validated by Lluch-Canut [24] in 1999 and later validated with a Mexican population of health sector workers by Aparicio et al. [41]. It is made up of six factors with a total of 39 items: personal satisfaction (PS; 8 items); prosocial attitude (PA; 5 items); self-control (SC; 5 items); autonomy (AU; 5 items); problem-solving and self-actualization (PSSA; 9 items); and interpersonal relationship skills (IR; 7 items). The instrument is answered on a 4-point Likert-type scale. For the positive items, the scale is always/almost always (4), frequently (3), sometimes (2) and never/almost never (1); for negative items, this scale is reversed. The positive items are 4, 5, 11, 15-18, 20-23, 25-29, 32, and 35-37; the negative items are 1-3, 6-10, 12-14, 19, 24, 30, 31, 33, 34, 38, and 39. In the validation carried out by Aparicio et al. [41], the percentage of variance explained was 43.4%, and Cronbach's α coefficient was 0.86 for the global instrument. These authors did not report the internal consistency obtained by factor. With the present population and prior to the psychometric analysis developed here, McDonald's omega (ω) coefficient per factor was PS = 0.82; PA = 0.58; SC = 0.84; AU = 0.83; PSSA = 0.82; IR = 0.73).
Table 2. Descriptive statistics for the items of the Positive Mental Health Questionnaire
|
Items |
Item content |
M |
SD |
S |
K |
|
|
Factor 1: Personal satisfaction |
|
|
|
|
|
4 |
I like myself as I am. |
3.51 |
0.72 |
-1.40 |
1.40 |
|
6 |
I feel like I am about to explode. |
3.30 |
0.72 |
-1.00 |
1.21 |
|
7 |
I find life to be boring and monotonous. |
3.61 |
0.68 |
-1.94 |
3.75 |
|
12 |
I view the future with pessimism. |
3.76 |
0.57 |
-2.70 |
7.94 |
|
14 |
I see myself as less important than those around me. |
3.76 |
0.59 |
-2.85 |
8.23 |
|
31 |
I feel inept and useless. |
3.85 |
0.51 |
-4.01 |
16.89 |
|
38 |
I feel unsatisfied with myself. |
3.40 |
0.98 |
-1.52 |
1.01 |
|
39 |
I feel unsatisfied with the way I look. |
3.38 |
0.82 |
-1.29 |
1.05 |
|
|
Factor 2: Prosocial attitude |
|
|
|
|
|
1 |
I find it especially difficult to accept others when their attitudes are different from mine. |
3.40 |
0.67 |
-1.02 |
1.31 |
|
3 |
I find it particularly difficult to listen to people tell me their problems. |
3.65 |
0.64 |
-2.16 |
5.25 |
|
23 |
I feel that I am someone to be trusted. |
3.86 |
0.45 |
-3.79 |
16.59 |
|
25 |
I consider the needs of others. |
3.23 |
0.84 |
-0.71 |
-0.54 |
|
37 |
I like to help others. |
3.65 |
0.62 |
-1.87 |
3.43 |
|
|
Factor 3: Self-control |
|
|
|
|
|
2 |
Problems often cause me to feel blocked. |
3.44 |
0.66 |
-1.20 |
2.01 |
|
5 |
I am able to control myself when I feel negative emotions. |
3.12 |
0.82 |
-0.52 |
-0.57 |
|
21 |
I am able to control myself when I have negative thoughts. |
3.28 |
0.80 |
-0.76 |
-0.41 |
|
22 |
I am able to maintain a high level of self-control in conflictive situations in my life. |
3.33 |
0.71 |
-0.62 |
-0.66 |
|
26 |
When I experience unpleasant external pressure I am able to maintain my personal balance. |
3.24 |
0.77 |
-0.79 |
0.14 |
|
|
Factor 4: Autonomy |
|
|
|
|
|
10 |
I worry a lot about what others think of me. |
3.32 |
0.77 |
-1.04 |
0.75 |
|
13 |
The opinions of others have a strong influence on me when I have to make decisions. |
3.49 |
0.67 |
-1.24 |
1.46 |
|
19 |
It troubles me when people criticize me. |
3.31 |
0.79 |
-0.96 |
0.28 |
|
33 |
I find it hard to hold my own opinions. |
3.57 |
0.71 |
-1.67 |
2.29 |
|
34 |
When I have to make big decisions I feel very unsure of myself. |
3.34 |
0.79 |
-1.25 |
1.31 |
|
|
Factor 5: Problem-solving and self-actualization |
|
|
|
|
|
15 |
I am able to make decisions on my own. |
3.68 |
0.69 |
-2.35 |
5.13 |
|
16 |
I try to look for the positive side when bad things happen to me. |
3.45 |
0.69 |
-0.96 |
0.10 |
|
17 |
I try to improve myself as a person. |
3.73 |
0.59 |
-2.27 |
4.92 |
|
27 |
When there are changes in my surroundings I try to adapt to them. |
3.60 |
0.63 |
-1.43 |
1.49 |
|
28 |
In the face of a problem I am able to ask for information. |
3.59 |
0.66 |
-1.52 |
1.71 |
|
29 |
I find changes in my daily routine to be stimulating. |
3.23 |
0.76 |
-0.52 |
-0.71 |
|
32 |
I try to develop my abilities to the maximum. |
3.61 |
0.64 |
-1.73 |
3.03 |
|
35 |
I am able to say no when I want to. |
3.15 |
0.92 |
-0.69 |
-0.67 |
|
36 |
When I am faced with a problem I try to find possible solutions. |
3.73 |
0.54 |
-2.22 |
5.62 |
|
|
Factor 6: Interpersonal relationship skills |
|
|
|
|
|
8 |
I find it particularly difficult to provide emotional support to others. |
3.48 |
0.71 |
-1.39 |
1.85 |
|
9 |
I find it hard to establish deep and satisfying interpersonal relationships with some people. |
3.50 |
0.71 |
-1.35 |
1.42 |
|
11 |
I feel that I have a strong ability to put myself in the shoes of others and to understand their responses. |
3.04 |
0.93 |
-0.57 |
-0.73 |
|
18 |
I consider myself to be a good professional. |
3.74 |
0.54 |
-2.34 |
6.25 |
|
20 |
I think that I am a sociable person. |
3.19 |
0.84 |
-0.67 |
-0.52 |
|
24 |
I find it particularly hard to understand the feelings of others. |
3.40 |
0.73 |
-1.10 |
0.88 |
|
30 |
I find it hard to relate openly with my teachers/bosses. |
3.35 |
0.83 |
-1.22 |
0.84 |
Abbreviations: K: kurtosis; M: mean; S: skewness; SD: standard deviation.
Major comments:
Table 2: Left align the second column. I think that “Item content” or “Item formulation” would be a more adequate column name.
Response:
The suggestion was heeded.
Change:
Table 2. Descriptive statistics for the items of the Positive Mental Health Questionnaire.
|
Items |
Item content |
M |
SD |
S |
K |
|
|
Factor 1: Personal satisfaction |
|
|
|
|
|
4 |
I like myself as I am. |
3.51 |
0.72 |
-1.40 |
1.40 |
|
6 |
I feel like I am about to explode. |
3.30 |
0.72 |
-1.00 |
1.21 |
|
7 |
I find life to be boring and monotonous. |
3.61 |
0.68 |
-1.94 |
3.75 |
|
12 |
I view the future with pessimism. |
3.76 |
0.57 |
-2.70 |
7.94 |
|
14 |
I see myself as less important than those around me. |
3.76 |
0.59 |
-2.85 |
8.23 |
|
31 |
I feel inept and useless. |
3.85 |
0.51 |
-4.01 |
16.89 |
|
38 |
I feel unsatisfied with myself. |
3.40 |
0.98 |
-1.52 |
1.01 |
|
39 |
I feel unsatisfied with the way I look. |
3.38 |
0.82 |
-1.29 |
1.05 |
|
|
Factor 2: Prosocial attitude |
|
|
|
|
|
1 |
I find it especially difficult to accept others when their attitudes are different from mine. |
3.40 |
0.67 |
-1.02 |
1.31 |
|
3 |
I find it particularly difficult to listen to people tell me their problems. |
3.65 |
0.64 |
-2.16 |
5.25 |
|
23 |
I feel that I am someone to be trusted. |
3.86 |
0.45 |
-3.79 |
16.59 |
|
25 |
I consider the needs of others. |
3.23 |
0.84 |
-0.71 |
-0.54 |
|
37 |
I like to help others. |
3.65 |
0.62 |
-1.87 |
3.43 |
|
|
Factor 3: Self-control |
|
|
|
|
|
2 |
Problems often cause me to feel blocked. |
3.44 |
0.66 |
-1.20 |
2.01 |
|
5 |
I am able to control myself when I feel negative emotions. |
3.12 |
0.82 |
-0.52 |
-0.57 |
|
21 |
I am able to control myself when I have negative thoughts. |
3.28 |
0.80 |
-0.76 |
-0.41 |
|
22 |
I am able to maintain a high level of self-control in conflictive situations in my life. |
3.33 |
0.71 |
-0.62 |
-0.66 |
|
26 |
When I experience unpleasant external pressure I am able to maintain my personal balance. |
3.24 |
0.77 |
-0.79 |
0.14 |
|
|
Factor 4: Autonomy |
|
|
|
|
|
10 |
I worry a lot about what others think of me. |
3.32 |
0.77 |
-1.04 |
0.75 |
|
13 |
The opinions of others have a strong influence on me when I have to make decisions. |
3.49 |
0.67 |
-1.24 |
1.46 |
|
19 |
It troubles me when people criticize me. |
3.31 |
0.79 |
-0.96 |
0.28 |
|
33 |
I find it hard to hold my own opinions. |
3.57 |
0.71 |
-1.67 |
2.29 |
|
34 |
When I have to make big decisions I feel very unsure of myself. |
3.34 |
0.79 |
-1.25 |
1.31 |
|
|
Factor 5: Problem-solving and self-actualization |
|
|
|
|
|
15 |
I am able to make decisions on my own. |
3.68 |
0.69 |
-2.35 |
5.13 |
|
16 |
I try to look for the positive side when bad things happen to me. |
3.45 |
0.69 |
-0.96 |
0.10 |
|
17 |
I try to improve myself as a person. |
3.73 |
0.59 |
-2.27 |
4.92 |
|
27 |
When there are changes in my surroundings I try to adapt to them. |
3.60 |
0.63 |
-1.43 |
1.49 |
|
28 |
In the face of a problem I am able to ask for information. |
3.59 |
0.66 |
-1.52 |
1.71 |
|
29 |
I find changes in my daily routine to be stimulating. |
3.23 |
0.76 |
-0.52 |
-0.71 |
|
32 |
I try to develop my abilities to the maximum. |
3.61 |
0.64 |
-1.73 |
3.03 |
|
35 |
I am able to say no when I want to. |
3.15 |
0.92 |
-0.69 |
-0.67 |
|
36 |
When I am faced with a problem I try to find possible solutions. |
3.73 |
0.54 |
-2.22 |
5.62 |
|
|
Factor 6: Interpersonal relationship skills |
|
|
|
|
|
8 |
I find it particularly difficult to provide emotional support to others. |
3.48 |
0.71 |
-1.39 |
1.85 |
|
9 |
I find it hard to establish deep and satisfying interpersonal relationships with some people. |
3.50 |
0.71 |
-1.35 |
1.42 |
|
11 |
I feel that I have a strong ability to put myself in the shoes of others and to understand their responses. |
3.04 |
0.93 |
-0.57 |
-0.73 |
|
18 |
I consider myself to be a good professional. |
3.74 |
0.54 |
-2.34 |
6.25 |
|
20 |
I think that I am a sociable person. |
3.19 |
0.84 |
-0.67 |
-0.52 |
|
24 |
I find it particularly hard to understand the feelings of others. |
3.40 |
0.73 |
-1.10 |
0.88 |
|
30 |
I find it hard to relate openly with my teachers/bosses. |
3.35 |
0.83 |
-1.22 |
0.84 |
Abbreviations: K: kurtosis; M: mean; S: skewness; SD: standard deviation.
Major comments:
Table 2: Also report the observed frequencies of the four item categories in the table. In addition, include the polychoric correlation matrix of all 39 items in a supplementary material or make your dataset publicly available.
Response:
The additional material presents the matrix of polychoric correlations that were estimated with the FACTOR V. 12.04.04 (WIN64) program.
Change:
Table 2A. Polychoric correlation matrix of the 39 items
|
|
PCDi |
V1 |
V2 |
V3 |
V4 |
V5 |
V6 |
V7 |
V8 |
V9 |
V10 |
V11 |
V12 |
V13 |
V14 |
V15 |
V16 |
V17 |
V18 |
V19 |
V20 |
V21 |
V22 |
V23 |
V24 |
V25 |
V26 |
V27 |
V28 |
V29 |
V30 |
V31 |
V32 |
V33 |
V34 |
V35 |
V36 |
V37 |
V38 |
V39 |
|
V1 |
7.90% |
-- -- |
0.339 |
0.056 |
-0.172 |
-0.086 |
0.416 |
-0.261 |
-0.278 |
-0.27 |
-0.231 |
0.053 |
0.281 |
0.222 |
0.223 |
0.182 |
0.193 |
0.092 |
0.098 |
0.318 |
0.364 |
-0.135 |
0.005 |
-0.142 |
0.246 |
0.248 |
0.308 |
0.314 |
0.267 |
0.165 |
0.244 |
-0.065 |
-0.305 |
-0.008 |
-0.192 |
-0.232 |
-0.169 |
-0.012 |
-0.006 |
0.034 |
|
V2 |
13.40% |
0.339 |
-- -- |
-0.053 |
-0.185 |
-0.179 |
0.206 |
-0.204 |
0.028 |
0.038 |
-0.062 |
0.038 |
0.215 |
0.23 |
0.198 |
0.081 |
0.265 |
0.069 |
0.051 |
0.511 |
0.376 |
-0.239 |
-0.018 |
-0.205 |
0.431 |
0.254 |
-0.055 |
0.093 |
0.068 |
0.178 |
-0.012 |
0.035 |
-0.128 |
0.027 |
-0.058 |
-0.045 |
0.041 |
-0.09 |
-0.102 |
0.006 |
|
V3 |
52.30% |
0.075 |
-0.071 |
-- -- |
0.281 |
0.255 |
0.085 |
0.11 |
0.283 |
0.355 |
0.286 |
0.135 |
-0.089 |
0.057 |
-0.066 |
-0.222 |
-0.256 |
0.023 |
-0.084 |
-0.105 |
0.061 |
0.157 |
0.471 |
0.174 |
-0.16 |
-0.088 |
0.063 |
-0.077 |
-0.129 |
-0.11 |
-0.103 |
0.217 |
0.208 |
0.272 |
0.428 |
0.417 |
0.232 |
0.326 |
0.232 |
0.347 |
|
V4 |
17.60% |
-0.172 |
-0.185 |
0.376 |
-- -- |
0.409 |
-0.103 |
0.171 |
0.257 |
0.192 |
0.396 |
0.079 |
-0.098 |
-0.216 |
0.036 |
0.028 |
0.033 |
-0.052 |
0.047 |
-0.268 |
-0.24 |
0.271 |
0.361 |
0.158 |
-0.257 |
-0.065 |
0.121 |
0.017 |
0.127 |
0.09 |
-0.076 |
0.177 |
0.312 |
0.266 |
0.431 |
0.325 |
0.233 |
0.442 |
0.091 |
0.232 |
|
V5 |
51.40% |
-0.115 |
-0.24 |
0.455 |
0.547 |
-- -- |
-0.108 |
0.112 |
0.146 |
0.172 |
0.281 |
0.123 |
-0.041 |
-0.101 |
-0.043 |
0.018 |
-0.034 |
-0.149 |
0.035 |
-0.226 |
-0.28 |
0.241 |
0.318 |
0.161 |
-0.355 |
-0.127 |
0.08 |
-0.096 |
0.039 |
-0.198 |
-0.02 |
0.189 |
0.309 |
0.267 |
0.353 |
0.342 |
0.237 |
0.375 |
0.169 |
0.311 |
|
V6 |
7.80% |
0.416 |
0.206 |
0.114 |
-0.103 |
-0.144 |
-- -- |
-0.461 |
-0.362 |
-0.427 |
-0.436 |
-0.275 |
0.304 |
0.306 |
0.469 |
0.456 |
0.385 |
0.319 |
0.322 |
0.334 |
0.414 |
-0.049 |
-0.217 |
-0.167 |
0.197 |
0.326 |
0.374 |
0.422 |
0.359 |
0.299 |
0.47 |
-0.165 |
-0.383 |
0.05 |
-0.321 |
-0.266 |
-0.27 |
-0.27 |
-0.179 |
-0.17 |
|
V7 |
8.20% |
-0.261 |
-0.204 |
0.146 |
0.171 |
0.149 |
-0.461 |
-- -- |
0.626 |
0.589 |
0.384 |
0.388 |
-0.214 |
-0.173 |
-0.266 |
-0.359 |
-0.197 |
-0.174 |
-0.219 |
-0.33 |
-0.345 |
0.237 |
0.256 |
0.113 |
-0.197 |
-0.129 |
-0.32 |
-0.349 |
-0.288 |
-0.283 |
-0.23 |
0.249 |
0.467 |
0.206 |
0.332 |
0.279 |
0.209 |
0.262 |
0.27 |
0.303 |
|
V8 |
10.80% |
-0.278 |
0.028 |
0.378 |
0.257 |
0.196 |
-0.362 |
0.626 |
-- -- |
0.744 |
0.594 |
0.373 |
-0.318 |
-0.261 |
-0.347 |
-0.337 |
-0.251 |
-0.163 |
-0.204 |
-0.206 |
-0.251 |
0.196 |
0.361 |
0.358 |
-0.083 |
-0.068 |
-0.328 |
-0.317 |
-0.4 |
-0.149 |
-0.344 |
0.369 |
0.624 |
0.358 |
0.517 |
0.45 |
0.441 |
0.489 |
0.288 |
0.377 |
|
V9 |
10.60% |
-0.27 |
0.038 |
0.475 |
0.192 |
0.23 |
-0.427 |
0.589 |
0.744 |
-- -- |
0.607 |
0.337 |
-0.278 |
-0.21 |
-0.258 |
-0.405 |
-0.241 |
-0.189 |
-0.246 |
-0.174 |
-0.216 |
0.246 |
0.423 |
0.22 |
-0.129 |
-0.104 |
-0.242 |
-0.284 |
-0.229 |
-0.311 |
-0.381 |
0.311 |
0.577 |
0.288 |
0.549 |
0.52 |
0.334 |
0.371 |
0.189 |
0.356 |
|
V10 |
10.20% |
-0.231 |
-0.062 |
0.382 |
0.396 |
0.376 |
-0.436 |
0.384 |
0.594 |
0.607 |
-- -- |
0.234 |
-0.214 |
-0.109 |
-0.289 |
-0.376 |
-0.295 |
-0.227 |
-0.224 |
-0.226 |
-0.258 |
0.161 |
0.473 |
0.299 |
-0.032 |
-0.189 |
-0.283 |
-0.345 |
-0.293 |
-0.267 |
-0.434 |
0.358 |
0.586 |
0.304 |
0.649 |
0.549 |
0.437 |
0.529 |
0.292 |
0.416 |
|
V11 |
9.00% |
0.053 |
0.038 |
0.18 |
0.079 |
0.164 |
-0.275 |
0.388 |
0.373 |
0.337 |
0.234 |
-- -- |
-0.149 |
-0.227 |
-0.54 |
-0.476 |
-0.278 |
-0.251 |
-0.445 |
-0.038 |
-0.333 |
0.108 |
0.337 |
0.25 |
-0.004 |
-0.213 |
-0.331 |
-0.309 |
-0.27 |
-0.329 |
-0.353 |
0.297 |
0.289 |
0.207 |
0.151 |
0.212 |
0.201 |
0.243 |
0.291 |
0.182 |
|
V12 |
48.10% |
0.376 |
0.287 |
-0.16 |
-0.13 |
-0.074 |
0.406 |
-0.286 |
-0.425 |
-0.371 |
-0.286 |
-0.199 |
-- -- |
0.237 |
0.452 |
0.356 |
0.202 |
0.175 |
0.21 |
0.256 |
0.195 |
-0.112 |
-0.102 |
-0.082 |
0.154 |
0.133 |
0.14 |
0.189 |
0.205 |
0.085 |
0.152 |
-0.087 |
-0.238 |
-0.118 |
-0.201 |
-0.143 |
-0.131 |
-0.117 |
-0.06 |
-0.105 |
|
V13 |
47.30% |
0.297 |
0.307 |
0.101 |
-0.288 |
-0.18 |
0.409 |
-0.231 |
-0.349 |
-0.28 |
-0.146 |
-0.304 |
0.423 |
-- -- |
0.453 |
0.221 |
0.199 |
0.201 |
0.223 |
0.288 |
0.294 |
-0.097 |
-0.04 |
-0.001 |
0.341 |
0.217 |
0.23 |
0.208 |
0.232 |
0.082 |
0.119 |
0.027 |
-0.212 |
-0.084 |
-0.125 |
-0.185 |
-0.198 |
-0.143 |
-0.079 |
0.019 |
|
V14 |
11.40% |
0.223 |
0.198 |
-0.089 |
0.036 |
-0.058 |
0.469 |
-0.266 |
-0.347 |
-0.258 |
-0.289 |
-0.54 |
0.604 |
0.606 |
-- -- |
0.694 |
0.512 |
0.51 |
0.452 |
0.359 |
0.507 |
-0.077 |
-0.421 |
-0.312 |
0.122 |
0.379 |
0.464 |
0.374 |
0.476 |
0.386 |
0.478 |
-0.231 |
-0.209 |
-0.195 |
-0.244 |
-0.224 |
-0.365 |
-0.279 |
-0.28 |
-0.16 |
|
V15 |
9.50% |
0.182 |
0.081 |
-0.296 |
0.028 |
0.024 |
0.456 |
-0.359 |
-0.337 |
-0.405 |
-0.376 |
-0.476 |
0.476 |
0.296 |
0.694 |
-- -- |
0.543 |
0.506 |
0.589 |
0.125 |
0.326 |
0.035 |
-0.514 |
-0.143 |
0.107 |
0.442 |
0.523 |
0.454 |
0.502 |
0.4 |
0.619 |
-0.265 |
-0.229 |
-0.163 |
-0.353 |
-0.334 |
-0.219 |
-0.382 |
-0.376 |
-0.305 |
|
V16 |
47.40% |
0.258 |
0.354 |
-0.457 |
0.044 |
-0.061 |
0.514 |
-0.263 |
-0.336 |
-0.322 |
-0.394 |
-0.371 |
0.361 |
0.355 |
0.685 |
0.726 |
-- -- |
0.494 |
0.434 |
0.061 |
0.141 |
0.077 |
-0.392 |
-0.216 |
0.003 |
0.475 |
0.28 |
0.209 |
0.272 |
0.254 |
0.375 |
-0.174 |
-0.226 |
-0.178 |
-0.329 |
-0.337 |
-0.204 |
-0.197 |
-0.357 |
-0.214 |
|
V17 |
11.30% |
0.092 |
0.069 |
0.031 |
-0.052 |
-0.2 |
0.319 |
-0.174 |
-0.163 |
-0.189 |
-0.227 |
-0.251 |
0.234 |
0.269 |
0.51 |
0.506 |
0.66 |
-- -- |
0.658 |
0.04 |
0.215 |
0.032 |
-0.282 |
-0.185 |
0.096 |
0.292 |
0.233 |
0.202 |
0.222 |
0.273 |
0.248 |
-0.029 |
-0.217 |
-0.179 |
-0.176 |
-0.094 |
-0.114 |
-0.24 |
-0.248 |
-0.143 |
|
V18 |
8.80% |
0.098 |
0.051 |
-0.112 |
0.047 |
0.047 |
0.322 |
-0.219 |
-0.204 |
-0.246 |
-0.224 |
-0.445 |
0.28 |
0.298 |
0.452 |
0.589 |
0.58 |
0.658 |
-- -- |
0.09 |
0.192 |
0.068 |
-0.222 |
-0.175 |
0.053 |
0.342 |
0.312 |
0.359 |
0.267 |
0.321 |
0.397 |
-0.091 |
-0.177 |
-0.126 |
-0.054 |
-0.248 |
-0.18 |
-0.168 |
-0.362 |
-0.205 |
|
V19 |
9.80% |
0.318 |
0.511 |
-0.141 |
-0.268 |
-0.303 |
0.334 |
-0.33 |
-0.206 |
-0.174 |
-0.226 |
-0.038 |
0.342 |
0.384 |
0.359 |
0.125 |
0.082 |
0.04 |
0.09 |
-- -- |
0.502 |
-0.14 |
-0.154 |
-0.245 |
0.421 |
0.249 |
0.129 |
0.17 |
0.154 |
0.224 |
0.146 |
-0.04 |
-0.245 |
-0.093 |
-0.119 |
-0.145 |
-0.037 |
-0.218 |
-0.016 |
-0.208 |
|
V20 |
6.50% |
0.364 |
0.376 |
0.081 |
-0.24 |
-0.374 |
0.414 |
-0.345 |
-0.251 |
-0.216 |
-0.258 |
-0.333 |
0.26 |
0.393 |
0.507 |
0.326 |
0.189 |
0.215 |
0.192 |
0.502 |
-- -- |
-0.168 |
-0.12 |
-0.462 |
0.415 |
0.44 |
0.322 |
0.202 |
0.26 |
0.228 |
0.348 |
0.05 |
-0.298 |
-0.117 |
-0.047 |
-0.214 |
-0.18 |
-0.149 |
-0.061 |
0.016 |
|
V21 |
51.70% |
-0.18 |
-0.32 |
0.28 |
0.362 |
0.431 |
-0.065 |
0.317 |
0.262 |
0.329 |
0.215 |
0.145 |
-0.199 |
-0.173 |
-0.103 |
0.047 |
0.137 |
0.043 |
0.091 |
-0.187 |
-0.224 |
-- -- |
0.23 |
0.16 |
-0.249 |
0 |
0.027 |
-0.014 |
0.002 |
-0.109 |
0.003 |
0.171 |
0.227 |
0.217 |
0.261 |
0.253 |
0.18 |
0.248 |
0.084 |
0.162 |
|
V22 |
14.40% |
0.005 |
-0.018 |
0.63 |
0.361 |
0.425 |
-0.217 |
0.256 |
0.361 |
0.423 |
0.473 |
0.337 |
-0.137 |
-0.053 |
-0.421 |
-0.514 |
-0.524 |
-0.282 |
-0.222 |
-0.154 |
-0.12 |
0.307 |
-- -- |
0.36 |
-0.06 |
-0.193 |
-0.239 |
-0.291 |
-0.311 |
-0.252 |
-0.3 |
0.342 |
0.4 |
0.361 |
0.549 |
0.489 |
0.467 |
0.588 |
0.28 |
0.376 |
|
V23 |
9.50% |
-0.142 |
-0.205 |
0.232 |
0.158 |
0.215 |
-0.167 |
0.113 |
0.358 |
0.22 |
0.299 |
0.25 |
-0.11 |
-0.001 |
-0.312 |
-0.143 |
-0.289 |
-0.185 |
-0.175 |
-0.245 |
-0.462 |
0.213 |
0.36 |
-- -- |
-0.058 |
-0.24 |
-0.102 |
-0.061 |
-0.248 |
0.017 |
-0.225 |
0.15 |
0.39 |
0.227 |
0.179 |
0.257 |
0.276 |
0.292 |
0.133 |
0.217 |
|
V24 |
16.30% |
0.246 |
0.431 |
-0.213 |
-0.257 |
-0.474 |
0.197 |
-0.197 |
-0.083 |
-0.129 |
-0.032 |
-0.004 |
0.206 |
0.456 |
0.122 |
0.107 |
0.004 |
0.096 |
0.053 |
0.421 |
0.415 |
-0.333 |
-0.06 |
-0.058 |
-- -- |
0.306 |
0.051 |
0.084 |
0.073 |
0.27 |
-0.006 |
0.015 |
-0.111 |
-0.069 |
-0.155 |
-0.154 |
-0.055 |
-0.184 |
0.006 |
-0.032 |
|
V25 |
11.40% |
0.248 |
0.254 |
-0.117 |
-0.065 |
-0.17 |
0.326 |
-0.129 |
-0.068 |
-0.104 |
-0.189 |
-0.213 |
0.178 |
0.29 |
0.379 |
0.442 |
0.635 |
0.292 |
0.342 |
0.249 |
0.44 |
0 |
-0.193 |
-0.24 |
0.306 |
-- -- |
0.245 |
0.227 |
0.328 |
0.309 |
0.368 |
-0.167 |
-0.089 |
0.042 |
-0.087 |
-0.309 |
-0.095 |
-0.058 |
-0.182 |
-0.186 |
|
V26 |
5.40% |
0.308 |
-0.055 |
0.084 |
0.121 |
0.107 |
0.374 |
-0.32 |
-0.328 |
-0.242 |
-0.283 |
-0.331 |
0.188 |
0.308 |
0.464 |
0.523 |
0.375 |
0.233 |
0.312 |
0.129 |
0.322 |
0.036 |
-0.239 |
-0.102 |
0.051 |
0.245 |
-- -- |
0.558 |
0.722 |
0.286 |
0.435 |
-0.154 |
-0.205 |
0.06 |
-0.175 |
-0.21 |
-0.229 |
-0.11 |
-0.327 |
-0.206 |
|
V27 |
4.70% |
0.314 |
0.093 |
-0.103 |
0.017 |
-0.128 |
0.422 |
-0.349 |
-0.317 |
-0.284 |
-0.345 |
-0.309 |
0.252 |
0.278 |
0.374 |
0.454 |
0.279 |
0.202 |
0.359 |
0.17 |
0.202 |
-0.019 |
-0.291 |
-0.061 |
0.084 |
0.227 |
0.558 |
-- -- |
0.532 |
0.374 |
0.458 |
-0.179 |
-0.22 |
0.036 |
-0.208 |
-0.22 |
-0.203 |
-0.174 |
-0.39 |
-0.132 |
|
V28 |
6.00% |
0.267 |
0.068 |
-0.172 |
0.127 |
0.052 |
0.359 |
-0.288 |
-0.4 |
-0.229 |
-0.293 |
-0.27 |
0.274 |
0.31 |
0.476 |
0.502 |
0.363 |
0.222 |
0.267 |
0.154 |
0.26 |
0.002 |
-0.311 |
-0.248 |
0.073 |
0.328 |
0.722 |
0.532 |
-- -- |
0.241 |
0.367 |
-0.161 |
-0.234 |
0.058 |
-0.255 |
-0.337 |
-0.397 |
-0.249 |
-0.293 |
-0.25 |
|
V29 |
8.60% |
0.165 |
0.178 |
-0.148 |
0.09 |
-0.265 |
0.299 |
-0.283 |
-0.149 |
-0.311 |
-0.267 |
-0.329 |
0.113 |
0.11 |
0.386 |
0.4 |
0.339 |
0.273 |
0.321 |
0.224 |
0.228 |
-0.145 |
-0.252 |
0.017 |
0.27 |
0.309 |
0.286 |
0.374 |
0.241 |
-- -- |
0.419 |
-0.259 |
-0.209 |
-0.08 |
-0.285 |
-0.333 |
-0.2 |
-0.21 |
-0.246 |
-0.289 |
|
V30 |
6.10% |
0.244 |
-0.012 |
-0.138 |
-0.076 |
-0.027 |
0.47 |
-0.23 |
-0.344 |
-0.381 |
-0.434 |
-0.353 |
0.204 |
0.16 |
0.478 |
0.619 |
0.501 |
0.248 |
0.397 |
0.146 |
0.348 |
0.004 |
-0.3 |
-0.225 |
-0.006 |
0.368 |
0.435 |
0.458 |
0.367 |
0.419 |
-- -- |
-0.239 |
-0.249 |
0.031 |
-0.34 |
-0.265 |
-0.265 |
-0.275 |
-0.286 |
-0.231 |
|
V31 |
49.80% |
-0.087 |
0.047 |
0.387 |
0.236 |
0.337 |
-0.221 |
0.332 |
0.494 |
0.415 |
0.478 |
0.396 |
-0.156 |
0.048 |
-0.309 |
-0.354 |
-0.311 |
-0.038 |
-0.121 |
-0.054 |
0.066 |
0.305 |
0.458 |
0.201 |
0.02 |
-0.223 |
-0.206 |
-0.24 |
-0.215 |
-0.347 |
-0.32 |
-- -- |
0.3 |
0.212 |
0.295 |
0.284 |
0.228 |
0.241 |
0.144 |
0.245 |
|
V32 |
12.40% |
-0.305 |
-0.128 |
0.278 |
0.312 |
0.413 |
-0.383 |
0.467 |
0.624 |
0.577 |
0.586 |
0.289 |
-0.318 |
-0.283 |
-0.209 |
-0.229 |
-0.302 |
-0.217 |
-0.177 |
-0.245 |
-0.298 |
0.303 |
0.4 |
0.39 |
-0.111 |
-0.089 |
-0.205 |
-0.22 |
-0.234 |
-0.209 |
-0.249 |
0.4 |
-- -- |
0.49 |
0.489 |
0.521 |
0.457 |
0.52 |
0.175 |
0.358 |
|
V33 |
50.90% |
-0.011 |
0.036 |
0.485 |
0.356 |
0.476 |
0.066 |
0.275 |
0.479 |
0.385 |
0.406 |
0.277 |
-0.21 |
-0.15 |
-0.261 |
-0.217 |
-0.319 |
-0.239 |
-0.168 |
-0.125 |
-0.157 |
0.387 |
0.482 |
0.304 |
-0.093 |
0.056 |
0.08 |
0.048 |
0.078 |
-0.106 |
0.042 |
0.379 |
0.654 |
-- -- |
0.382 |
0.335 |
0.359 |
0.409 |
0.104 |
0.241 |
|
V34 |
14.30% |
-0.192 |
-0.058 |
0.571 |
0.431 |
0.471 |
-0.321 |
0.332 |
0.517 |
0.549 |
0.649 |
0.151 |
-0.268 |
-0.167 |
-0.244 |
-0.353 |
-0.44 |
-0.176 |
-0.054 |
-0.119 |
-0.047 |
0.349 |
0.549 |
0.179 |
-0.155 |
-0.087 |
-0.175 |
-0.208 |
-0.255 |
-0.285 |
-0.34 |
0.394 |
0.489 |
0.511 |
-- -- |
0.649 |
0.503 |
0.557 |
0.288 |
0.476 |
|
V35 |
13.70% |
-0.232 |
-0.045 |
0.558 |
0.325 |
0.457 |
-0.266 |
0.279 |
0.45 |
0.52 |
0.549 |
0.212 |
-0.191 |
-0.247 |
-0.224 |
-0.334 |
-0.451 |
-0.094 |
-0.248 |
-0.145 |
-0.214 |
0.337 |
0.489 |
0.257 |
-0.154 |
-0.309 |
-0.21 |
-0.22 |
-0.337 |
-0.333 |
-0.265 |
0.379 |
0.521 |
0.448 |
0.649 |
-- -- |
0.515 |
0.438 |
0.294 |
0.4 |
|
V36 |
11.30% |
-0.169 |
0.041 |
0.31 |
0.233 |
0.316 |
-0.27 |
0.209 |
0.441 |
0.334 |
0.437 |
0.201 |
-0.175 |
-0.265 |
-0.365 |
-0.219 |
-0.272 |
-0.114 |
-0.18 |
-0.037 |
-0.18 |
0.24 |
0.467 |
0.276 |
-0.055 |
-0.095 |
-0.229 |
-0.203 |
-0.397 |
-0.2 |
-0.265 |
0.304 |
0.457 |
0.48 |
0.503 |
0.515 |
-- -- |
0.526 |
0.164 |
0.318 |
|
V37 |
13.50% |
-0.012 |
-0.09 |
0.435 |
0.442 |
0.502 |
-0.27 |
0.262 |
0.489 |
0.371 |
0.529 |
0.243 |
-0.156 |
-0.191 |
-0.279 |
-0.382 |
-0.263 |
-0.24 |
-0.168 |
-0.218 |
-0.149 |
0.331 |
0.588 |
0.292 |
-0.184 |
-0.058 |
-0.11 |
-0.174 |
-0.249 |
-0.21 |
-0.275 |
0.322 |
0.52 |
0.547 |
0.557 |
0.438 |
0.526 |
-- -- |
0.22 |
0.439 |
|
V38 |
11.70% |
-0.006 |
-0.102 |
0.31 |
0.091 |
0.226 |
-0.179 |
0.27 |
0.288 |
0.189 |
0.292 |
0.291 |
-0.08 |
-0.106 |
-0.28 |
-0.376 |
-0.477 |
-0.248 |
-0.362 |
-0.016 |
-0.061 |
0.112 |
0.28 |
0.133 |
0.006 |
-0.182 |
-0.327 |
-0.39 |
-0.293 |
-0.246 |
-0.286 |
0.193 |
0.175 |
0.139 |
0.288 |
0.294 |
0.164 |
0.22 |
-- -- |
0.301 |
|
V39 |
50.10% |
0.045 |
0.008 |
0.62 |
0.31 |
0.555 |
-0.228 |
0.405 |
0.504 |
0.476 |
0.556 |
0.243 |
-0.187 |
0.033 |
-0.214 |
-0.408 |
-0.382 |
-0.192 |
-0.274 |
-0.278 |
0.022 |
0.29 |
0.503 |
0.29 |
-0.043 |
-0.248 |
-0.276 |
-0.177 |
-0.334 |
-0.387 |
-0.309 |
0.437 |
0.479 |
0.43 |
0.636 |
0.534 |
0.424 |
0.586 |
0.402 |
-- -- |
PCDi: Percentage of covariance destroyed in each variable
Major comments:
Sect. 3.4.: You repeatedly argue that you cannot rely on the six-factor solution because the covariance matrix is “positive defined.” First, the correct term is “positive definite”. Second, I suppose the warning is related to the covariance matrix of parameter estimates (i.e., related standard errors). Issues can be circumvented by using constrained estimation. It could also be the case that you have sparse data (i.e., you have no or only a few observations in some categories for some items). These data issues must not dictate the choice of your final model. That is, stick to your six-factor model and investigate the cause of the reported issues. Moreover, you reason that some factors are “highly correlated” and should be merged. However, the reported factor correlations do not exceed typical cutoffs for high correlations (e.g.,.85). Hence, using fewer factors is unjustified in your case.
Response:
The term “positive defined” was replaced by “positive definite”. Likewise, by constrained estimation between factorswith high correlation (i.e., HR ~~ AP and RP ~~ AU), the correlation matrix was positive definite. Therefore, the 6-factoranalysis was continued. The observation is appreciated. However, the 4-factor and factor models were also estimated for theoretical and empirical reasons.
Change:
3.3.1. Original Structure with 6 Factors
Confirmatory factor analysis of the original version with 6 factors of the PMHQ did not show a positive definite covariance matrix. Table 3 (without constraints) shows the correlation matrix between the six factors and reveals a high correlation for Factor 3 with Factor 5 and for Factor 2 and Factor 6. Additionally, an eigenvalue was negative (i.e., -0.0007). A covariation constraint for both pairs of factors was fixed at 0.05, and the model was estimated again. The covariance matrix was now positive definite with all eigenvalues > 0. The correlation matrix between factors is shown in Table 3 (with constraints). All factor loadings were significant (p < 0.001). The standardized λ ranged between 0.12 and 0.84 (see Table 2B in Supplementary Materials Section), and all covariances between factors were significant (range for standardized values = 0.16 to 0.79, p < 0.05). The goodness-of-fit indices and mean factor loading for this structure are shown in Table 4. Except for the χ2 test and WRMR, the rest were acceptable. All Mλ were < 0.70.
1.5. Research Questions
The studies that have reported high correlations between PMHQ factors (e.g., [7,45,46,52]) have used exploratory factor analysis, which is a data-driven approach that lacks indices that allow evaluation of the goodness-of-fit of the models (Prokofieva et al., 2023). Confirmatory factor analysis, a theory-driven approach, can inflate the factor correlations and produce biased goodness-of-fit indices due to ignoring cross-loading (Alamer, 2022). When the correlation between factors ranges between 0.80 and 0.85, there is a risk of overfactoring the structure, which compromises its discriminant validity (Watkins, 2023). These facts motivated the first research question:
- Will the original structure of the PMHQ, based on previous studies (e.g., [7,45,46,52]) that reported high correlations between its factors, show poor fit?
Derived from the above, it is possible that a theory-and-data-driven modification of the PMHQ, based on the conceptual analysis of correlated factors, will allow the identification of a model with a better fit than the original structure. This motivated the second research question:
- Can a PMHQ structure with fewer than 6 factors (original structure) but more than 1 demonstrate an acceptable fit?
Finally, Cabarcas and Mendoza [42], through an exploratory factor analysis, detected that in the original 6-factor structure of the PMHQ, the first factor explained 67.18% of the variance. The remaining factors explained a value of less than 5%. This led them to evaluate a one-factor structure for this instrument. This motivated the third and last research question:
- Will a unifactorial structure of the PMHQ yield an acceptable fit?
In this context, the primary objective of this study was to evaluate the psychometric properties of the PMHQ from a structural equations approach, that is, by estimating its structure through CFA. A secondary objective was to evaluate the structure of this instrument with 1 factor or a smaller number of factors compared to the original structure but greater than 1.
Major comments:
Using ordinal alpha for ordinal items is an inappropriate reliability measure. Please consult Chalmers (2018), Green and Yang (2009), and https://github.com/simsem/semTools/issues/48. Remove anything related to ordinal alpha in the manuscript. Use a proper reliability statistic for ordinal data (see the discussion in Github in the link above).
Response:
Any reference to ordinal alpha was eliminated, and McDonald's omega was used to estimate internal consistency.
Change:
2.6.4. Internal Consistency
Internal consistency was evaluated using Cronbach's alpha (α) to compare studies, and McDonald's ω for estimating reliability more accurately than α [66]. For Cronbach's α coefficient, a value ≥ .70 was considered acceptable [67], and for McDonald's ω, a value ≥ .80 (Viladrich, et al., 2017) and ≤ .94 (Kline, 2015) were considered acceptable.
3.4. Internal Consistency
Table 6 shows the values for Cronbach's alpha and McDonald's omega for the structures from the PMHQ with six and four factors and one factor. Based only on McDonald's omega, for the six-factor structure, the internal consistency was acceptable only for Factor 1 and Factor 3 to Factor 5. For the four-factor structure, all factors but Factor 2 were acceptable. The one-factor structure was also acceptable.
Table 6. Internal consistency of the structures evaluated in the Positive Mental Health Questionnaire
|
Factors |
F1 |
F2 |
F3 |
F4 |
F5 |
F6 |
|
6 |
α = 0.75; ω = 0.82 |
α = 0.47; ω = 0.58 |
α = 0.78; ω = 0.84 |
α = 0.69; ω = 0.83 |
α = 0.77; ω = 0.82 |
α = 0.59; ω = 0.73 |
|
4 |
α = 0.75; ω = 0.82 |
α = 0.70; ω = 0.75 |
α = 0.85; ω = 0.87 |
α = 0.69; ω = 0.83 |
|
|
|
1 |
α = 0.89; ω = 0.90 |
|
|
|
|
|
Abbreviations: α: Cronbach’s alpha; ω: McDonald’s omega.
4.2. Internal Consistency of the One-factor, Four-factor and Six-factor Positive Mental Health Questionnaire
For the structure with 6 factors, from the calculation of McDonald's omega, an acceptable internal consistency was found in 4 of them. In the case of the model with 4 factors, there was an acceptable internal consistency in 3 of them, which indicates that the results obtained from these factors are repeatable and consistent. Factors with unacceptable internal consistency suggest that the items are heterogeneous depending on what they claim to measure, since the omega is based on a factor analysis model that examines the interrelationships between elements and subsets of elements (Ferketich, 1990). Now, the fact that factor 2 increases its internal consistency in the model with 4 factors supports the idea of uniting factors 6 and 2, which can be supported by the relationship between emotional competence, interpersonal relationships and prosocial behaviors, as detected in a study by Pung et al. (2021), which also points out that emotional competence and interpersonal relationships with peers promote psychosocial behavior.
With respect to the Cronbach's alpha obtained in previous studies in the case of the model with 6 factors, a systematically lower value in factor 2 is notable [14, 17-19]. This again justifies the union of the factor related to interpersonal relationships and prosocial behaviors in which the first acts as a mediator of the second; it is also based on the relationship between prosocial behaviors and variables existing in positive interpersonal behaviors such as empathy, desire for adventure, positive self-concept and self-esteem (Calvo et al., 2001).
Minor comments:
Abstract, Sect. 2.1.: Write M=37.06 and SD=10.79 (i.e., do not write mathematical symbols in parenthesis).
Abstract: Write “one-factor” instead of “unifactorial” structure.
Response:
The change has been made.
Change:
Abstract: The Positive Mental Health Questionnaire (PMHQ) has been validated across various populations but has displayed diverse psychometric structures depending on the procedures used. The original version of the PMHQ includes 39 items organized into 6 factors, although there are reports that indicate a reduced structure of between 1 and 4 factors. The aim of this study was to assess the psychometric properties of the PMHQ with 1, 4 and 6 factors. A total of 360 healthcare workers aged 23 to 77 (M = 37.06; SD = 10.79) participated. Construct validity was assessed through confirmatory factor analysis using weighted root mean square residual. The original 6-factor (χ2/df: 3.40; RMSEA: 0.085; CFI: 0.913; TLI: 0.906) and a reduced 4-factor (χ2/df: 2.90; RMSEA: 0.072; CFI: 0.931; TLI: 0.926) structure showed acceptable fit. The fit of the one-factor model was unacceptable. The internal consistency was evaluated through McDonald's ω, and it was acceptable for 4 of 6 factors of the original structure and for 3 of 4 factors of the reduced structure. In conclusion, these findings suggest that the 6-factor and 4-factor models are valid for measuring positive mental health. However, issues with internal consistency must be investigated.
Minor comments:
Sect. 2.2.: Is it better to say that Likert-scale items with “4 points” instead of “4 levels” are used?
Response:
The change has been made.
Change:
2.2.2. Positive Mental Health Questionnaire
The PMHQ was originally designed and validated by Lluch-Canut [24] in 1999 and later validated with a Mexican population of health sector workers by Aparicio et al. [41]. It is made up of six factors with a total of 39 items: personal satisfaction (PS; 8 items); prosocial attitude (PA; 5 items); self-control (SC; 5 items); autonomy (AU; 5 items); problem-solving and self-actualization (PSSA; 9 items); and interpersonal relationship skills (IR; 7 items). The instrument is measured on a 4-point Likert-type scale. For the positive items, the scale is always/almost always (4), frequently (3), sometimes (2) and never/almost never (1); for negative items, this scale is reversed. The positive items are 4, 5, 11, 15-18, 20-23, 25-29, 32, and 35-37. The negative items are 1-3, 6-10, 12-14, 19, 24, 30, 31, 33, 34, 38, and 39. In the validation carried out by Aparicio et al. [41], the percentage of variance explained was 43.4%, and Cronbach's α coefficient was 0.86 for the global instrument. These authors did not report the internal consistency obtained by factor. With the present population and prior to the psychometric analysis developed here, McDonald's omega (ω) coefficient per-factor was PS = 0.82; PA = 0.58; SC = 0.84; AU = 0.83; PSSA = 0.82; IR = 0.73.
Minor comments:
Sect. 2.6.4.: Do not write lambda in parenthesis.
Response:
The change has been made.
Change:
2.6.3. Confirmatory Factor Analysis
The construct validity of the PMHQ was assessed using CFA with the weighted least squares means and variance adjusted (WLSMV) estimator with data from the polychoric correlation and asymptotic covariance matrix [62, 63]. Items with p < 0.05 were retained, and whether they met the criterion of mean factor loading ≥.70 per factor was verified [64]. The correlations between factor scores generated from CFA were reported. The fit of the model was evaluated using the χ2 test, χ2/df, root mean square error of approximation (RMSEA), weighted root mean square residual (WRMR), comparative fit index (CFI) and Tucker‒Lewis index (TLI). The following fit values were considered acceptable: χ2/df ≤ 5, RMSEA ≤ 0.08 (90% CI < 0.10), WRMR ≤ 1, CFI ≥ 0.90, TLI ≥ 0.90; and excellent fit: χ2/df ≤ 2, RMSEA ≤ 0.05, CFI ≥ 0.95, TLI ≥ 0.95 [65]. Goodness-of-fit indices were evaluated sequentially, and in the case of values below what was acceptable, appropriate respecifications were made to the model. For each respecification, statistical criteria (modification indices and factorial saturation of each item) and theoretical criteria (item-factor and factor-construct conceptual coherence) were considered to maintain the conceptual value of the instrument.
Minor comments:
Sect. 3.1.: Remove the information about “data points.”
Response:
The change has been made.
Change:
3.1. Data processing
In a database with 360 observations for 39 variables, 34 missing data points were found, corresponding to 0.2% of the total (see Figure 1.A in the Supplementary Materials Section). There were no missing data completely at random, as Little´s test revealed (χ2 = 782, p = 0.03); therefore, each observation with at least one missing data point was removed from the dataset. This eliminates 24 observations and leaves n = 336 for posterior analysis. After this, 22 multivariate outliers were identified. However, visual inspection of the cases did not reveal the presence of systematic patterns of mechanical response, so the observations were kept for further analysis.
Minor comments:
Sect. 3.2.: Write “p < .01”.
Response:
The change has been made.
Change:
3.2. Descriptive Analysis, Univariate and Multivariate Normality and Polychoric Correlation
Table 2 presents the descriptive analyses of the items of the PMHQ. There was no evidence of univariate (Anderson‒Darling test, minimum value = 22.19 and maximum value = 101.29, p < 0.01) or multivariate normality (Mardia skewness = 24045.96; Mardia kurtosis = 53.83, p < 0.01). In the Supplementary Materials Section, the polychoric correlation matrix of all 39 items is shown (Table 2A).
Reviewer 2 Report
Comments and Suggestions for Authors
Reviewer Comments
September 8, 2023
Abstract Section:
· The content of the abstract is logical, detailing the research's purpose, methodology, results, and conclusion. However, there are areas for improvement in terms of language fluency and clarity of expression. More specifically,
o There are instances of redundancy and verbosity in the choice of words and phrasing. For example, “has been validated in diverse populations, but it has shown diverse psychometric structures for the variety of procedures used” can be streamlined to "has been validated across various populations but has displayed diverse psychometric structures depending on the procedures used."
o It's essential to maintain consistency and clarity when using technical terms and metrics. For instance, when mentioning multiple fit indices, ensure each is described concisely and clearly.
o The statement “The results showed that the original structure of the data correlation matrix was not positively defined” may be ambiguous for those unfamiliar with the topic. A clearer description of the issues with the original structure of the data correlation matrix might be necessary.
o The conclusion section can be further simplified for directness.
o An alternative version could be as follows for the authors’ reference: Abstract: The Positive Mental Health Questionnaire (PMHQ) has been validated across various populations but has displayed diverse psychometric structures depending on the procedures used. The aim of this study was to assess the psychometric properties of the PMHQ using a structural equation approach. 360 healthcare workers aged 23 to 77 (mean (M) = 37.06; standard deviation (SD) = 10.79), participated. Construct validity was assessed through confirmatory factor analysis. Model fit was evaluated with indicators like χ2, χ2/gl, root mean square error of approximation, weighted root means square residual, comparative fit index, and the Tucker‒Lewis index. Internal consistency was checked using Cronbach's α and the ordinal α. Results indicated issues with the original structure of the data correlation matrix, leading to inconsistent adjustment values. A reduced 4-factor structure showed acceptable fit and consistency. A unifactorial structure also had an acceptable fit and consistency but retained only 30 items. In conclusion, the four-factor model demonstrated the best fit and overall internal consistency, ranging from acceptable to improve per factor.
Introduction Section:
· 1.5. Problem Statement should be replaced with “Research Questions”. The following text could be an alternative version: To further understand the psychometric properties of the PMHQ, this study poses the following research questions based on structural equations approach:
· 1. Will the original structure of the PMHQ, based on previous studies (e.g., [7,45,46,52]) that reported high correlations between its factors, show poor fit?
· 2. Can a PMHQ structure with fewer than 6 factors (original structure) but more than 1 demonstrate an acceptable fit?
· 3. Will a unifactorial structure of the PMHQ yield an acceptable fit?
Materials and Methods Section:
· How did the authors justify the sample size in the present study?
· Did the researchers offer incentives to the participants? This aspect should be transparent to the readers.
· Women and men in Table 1 should be replaced with “Female” and “Male”, respectively.
· Table 1 should just have 3 lines: the 1st, 2nd, and the last line.
· Table 1 should have 3 columns: the 1st column (Variables), 2nd column (Category), and 3rd column (n, %).
· The authors do not have reason to report Cronbach’s alpha coefficients in the present study because you never tested whether the tau-equivalence assumption was satisfied. You can use omega coefficients instead.
· The authors should check if common method biases exist when running measurement models.
· Before running the SEM model, the authors should check outliers, normality etc. of the items used in the present study.
· How did the authors convert the latent variables into observed variables? Factor score?
· Please remove Figure 1 and Figure 2, respectively.
Comments on the Quality of English LanguageThe authors' English writing skills are OK.
Author Response
Comments and Suggestions for Authors
Comment
Abstract Section:
- The content of the abstract is logical, detailing the research's purpose, methodology, results, and conclusion. However, there are areas for improvement in terms of language fluency and clarity of expression. More specifically,
o There are instances of redundancy and verbosity in the choice of words and phrasing. For example, “has been validated in diverse populations, but it has shown diverse psychometric structures for the variety of procedures used” can be streamlined to "has been validated across various populations but has displayed diverse psychometric structures depending on the procedures used."
o It's essential to maintain consistency and clarity when using technical terms and metrics. For instance, when mentioning multiple fit indices, ensure each is described concisely and clearly.
o The statement “The results showed that the original structure of the data correlation matrix was not positively defined” may be ambiguous for those unfamiliar with the topic. A clearer description of the issues with the original structure of the data correlation matrix might be necessary.
o The conclusion section can be further simplified for directness.
An alternative version could be as follows for the authors’ reference: Abstract: The Positive Mental Health Questionnaire (PMHQ) has been validated across various populations but has displayed diverse psychometric structures depending on the procedures used. The aim of this study was to assess the psychometric properties of the PMHQ using a structural equation approach. 360 healthcare workers aged 23 to 77 (mean (M) = 37.06; standard deviation (SD) = 10.79), participated. Construct validity was assessed through confirmatory factor analysis. Model fit was evaluated with indicators like χ2, χ2/gl, root mean square error of approximation, weighted root means square residual, comparative fit index, and the Tucker‒Lewis index. Internal consistency was checked using Cronbach's α and the ordinal α. Results indicated issues with the original structure of the data correlation matrix, leading to inconsistent adjustment values. A reduced 4-factor structure showed acceptable fit and consistency. A unifactorial structure also had an acceptable fit and consistency but retained only 30 items. In conclusion, the four-factor model demonstrated the best fit and overall internal consistency, ranging from acceptable to improve per factor.
Response:
A new Abstract was prepared, with greater attention to content and writing.
Change:
Abstract: The Positive Mental Health Questionnaire (PMHQ) has been validated across various populations but has displayed diverse psychometric structures depending on the procedures used. The original version of the PMHQ includes 39 items organized into 6 factors, although there are reports that indicate a reduced structure of between 1 and 4 factors. The aim of this study was to assess the psychometric properties of the PMHQ with 1, 4 and 6 factors. A total of 360 healthcare workers aged 23 to 77 (M = 37.06; SD = 10.79) participated. Construct validity was assessed through confirmatory factor analysis using weighted root mean square residual. The original 6-factor (χ2/df: 3.40; RMSEA: 0.085; CFI: 0.913; TLI: 0.906) and a reduced 4-factor (χ2/df: 2.90; RMSEA: 0.072; CFI: 0.931; TLI: 0.926) structure showed acceptable fit. The fit of the one-factor model was unacceptable. The internal consistency was evaluated through McDonald's ω, and it was acceptable for 4 of 6 factors of the original structure and for 3 of 4 factors of the reduced structure. In conclusion, these findings suggest that the 6-factor and 4-factor models are valid for measuring positive mental health. However, issues with internal consistency must be investigated.
Comment
Introduction Section:
- 1.5. Problem Statement should be replaced with “Research Questions”.
Respuesta: Se realize el cambio
The following text could be an alternative version: To further understand the psychometric properties of the PMHQ, this study poses the following research questions based on structural equations approach:
- 1. Will the original structure of the PMHQ, based on previous studies (e.g., [7,45,46,52]) that reported high correlations between its factors, show poor fit?
- 2. Can a PMHQ structure with fewer than 6 factors (original structure) but more than 1 demonstrate an acceptable fit?
- 3. Will a unifactorial structure of the PMHQ yield an acceptable fit?
Respuesta: Se hizo el cambio y se agradece la sugerencia.
Response:
The change has been made.
Change:
1.5. Research Questions
The studies that have reported high correlations between PMHQ factors (e.g., [7,45,46,52]) have used exploratory factor analysis, which is a data-driven approach that lacks indices that allow evaluation of the goodness-of-fit of the models (Prokofieva et al., 2023). Confirmatory factor analysis, a theory-driven approach, can inflate the factor correlations and produce biased goodness-of-fit indices due to ignoring cross-loading (Alamer, 2022). When the correlation between factors ranges between 0.80 and 0.85, there is a risk of overfactoring the structure, which compromises its discriminant validity (Watkins, 2023). These facts motivated the first research question:
- Will the original structure of the PMHQ, based on previous studies (e.g., [7,45,46,52]) that reported high correlations between its factors, show poor fit?
Derived from the above, it is possible that a theory-and-data-driven modification of the PMHQ, based on the conceptual analysis of correlated factors, will allow the identification of a model with a better fit than the original structure. This motivated the second research question:
- Can a PMHQ structure with fewer than 6 factors (original structure) but more than 1 demonstrate an acceptable fit?
Finally, Cabarcas and Mendoza [42], through an exploratory factor analysis, detected that in the original 6-factor structure of the PMHQ, the first factor explained 67.18% of the variance. The remaining factors explained a value of less than 5%. This led them to evaluate a one-factor structure for this instrument. This motivated the third and last research question:
- Will a unifactorial structure of the PMHQ yield an acceptable fit?
In this context, the primary objective of this study was to evaluate the psychometric properties of the PMHQ from a structural equations approach, that is, by estimating its structure through CFA. A secondary objective was to evaluate the structure of this instrument with 1 factor or a smaller number of factors compared to the original structure but greater than 1.
Comment
Materials and Methods Section:
How did the authors justify the sample size in the present study?
Response:
The sample size was justified in section 2.1 by mentioning the minimum sample size recommended to satisfy the criteria of statistical power, effect size and significance level for studies with structural equation models.
Change:
2.1. Participants
Using a nonprobabilistic convenience sampling technique, a sample of 360 healthcare workers between the ages of 23 and 77 (mean (M) = 37.06; standard deviation (SD) = 10.79) with years of work experience between 0 and 53 years (M = 9.66) was recruited. The sample size is greater than the recommended minimum sample size for studies that use structural equation models, considering an anticipated effect size = 0.5, a statistical power level = 0.8 and probability level = 0.05 for a model with 6 latent variables and 39 observed variable switches equal to n = 288 (Soper, 2020). The inclusion criteria were workers in the medical or nursing area of the Federico Gómez Children's Hospital of Mexico who agreed to voluntarily participate in this study. The only exclusion criteria were not responding to three or more items of the PMHQ or closing the online form with the instruments, which caused the responses not to be recorded. Other sociodemographic and employment data are shown in Table 1. No type of material, economic or social incentive was offered to the participants for their collaboration in this study.
Comment
Did the researchers offer incentives to the participants? This aspect should be transparent to the readers.
Response:
Explicit clarification was made in section 2.1
Change:
2.1. Participants
Using a nonprobabilistic convenience sampling technique, a sample of 360 healthcare workers between the ages of 23 and 77 (mean (M) = 37.06; standard deviation (SD) = 10.79) with years of work experience between 0 and 53 years (M = 9.66) was recruited. The sample size is greater than the recommended minimum sample size for studies that use structural equation models, considering an anticipated effect size = 0.5, a statistical power level = 0.8 and probability level = 0.05 for a model with 6 latent variables and 39 observed variable switches equal to n = 288 (Soper, 2020). The inclusion criteria were workers in the medical or nursing area of the Federico Gómez Children's Hospital of Mexico who agreed to voluntarily participate in this study. The only exclusion criteria were not responding to three or more items of the PMHQ or closing the online form with the instruments, which caused the responses not to be recorded. Other sociodemographic and employment data are shown in Table 1. No type of material, economic or social incentive was offered to the participants for their collaboration in this study.
Comment
Women and men in Table 1 should be replaced with “Female” and “Male”, respectively.
Table 1 should just have 3 lines: the 1st, 2nd, and the last line.
Table 1 should have 3 columns: the 1st column (Variables), 2nd column (Category), and 3rd column (n, %).
Response:
The change has been made.
Change:
Table 1. Participant data for the sociodemographic variables questionnaire.
|
Variables |
Category |
n (%) |
|
Sex |
|
|
|
|
Female |
273 (75.8) |
|
|
Male |
86 (23.9) |
|
|
Missing |
1 (0.3) |
|
Civil status |
|
|
|
|
Single |
217 (60.3) |
|
|
Married |
86 (23.9) |
|
|
Consensual union |
40 (11.1) |
|
|
Divorced |
9 (2.5) |
|
|
Separated |
3 (0.8) |
|
|
Widowed |
5 (1.4) |
|
Educational level |
|
|
|
|
Technical school |
14 (3.9) |
|
|
Bachelor’s degree |
121 (33.6) |
|
|
Specialty |
110 (30.6) |
|
|
Subspecialty |
89 (24.7) |
|
|
Master’s degree |
22 (6.1) |
|
|
PhD |
3 (0.8) |
|
|
Missing |
1 (0.3) |
|
Area |
|
|
|
|
Medical |
237 (65.8) |
|
|
Nursing |
123 (34.2) |
|
Shift |
|
|
|
|
Morning |
178 (49.4) |
|
|
Evening |
25 (6.9) |
|
|
Night |
30 (8.3) |
|
|
Other |
127 (35.3) |
|
Second job in public sector |
|
|
|
|
Yes |
31 (8.6) |
|
|
No |
329 (91.4) |
|
Second job in private practice |
|
|
|
|
Yes |
49 (13.6) |
|
|
No |
311 (86.4) |
|
Diagnosis of psychopathology in the last 12 months |
|
|
|
|
Yes |
62 (17.2) |
|
|
No |
298 (82.8) |
|
Use of psychotropic drugs in the last 12 months |
|
|
|
|
Yes |
68 (18.9) |
|
|
No |
292 (81.1) |
Comment
The authors do not have reason to report Cronbach’s alpha coefficients in the present study because you never tested whether the tau-equivalence assumption was satisfied. You can use omega coefficients instead.
Response:
McDonald's omega was reported, and Cronbach's alpha was used only for comparative purposes with other studies.
Change:
2.6.4. Internal Consistency
Internal consistency was evaluated using Cronbach's alpha (α) to compare studies, and McDonald's ω for estimating reliability more accurately than α [66]. For Cronbach's α coefficient, a value ≥ .70 was considered acceptable [67], and for McDonald's ω, a value ≥ .80 (Viladrich, et al., 2017) and ≤ .94 (Kline, 2015) were considered acceptable.
3.4. Internal Consistency
Table 6 shows the values for Cronbach's alpha and McDonald's omega for the structures from the PMHQ with six and four factors and one factor. Based only on McDonald's omega, for the six-factor structure, the internal consistency was acceptable only for Factor 1 and Factor 3 to Factor 5. For the four-factor structure, all factors but Factor 2 were acceptable. The one-factor structure was also acceptable.
Table 6. Internal consistency of the structures evaluated in the Positive Mental Health Questionnaire
|
Factors |
F1 |
F2 |
F3 |
F4 |
F5 |
F6 |
|
6 |
α = 0.75; ω = 0.82 |
α = 0.47; ω = 0.58 |
α = 0.78; ω = 0.84 |
α = 0.69; ω = 0.83 |
α = 0.77; ω = 0.82 |
α = 0.59; ω = 0.73 |
|
4 |
α = 0.75; ω = 0.82 |
α = 0.70; ω = 0.75 |
α = 0.85; ω = 0.87 |
α = 0.69; ω = 0.83 |
|
|
|
1 |
α = 0.89; ω = 0.90 |
|
|
|
|
|
Abbreviations: α: Cronbach’s alpha; ω: McDonald’s omega.
4.2. Internal Consistency of the One-factor, Four-factor and Six-factor Positive Mental Health Questionnaire
For the structure with 6 factors, from the calculation of McDonald's omega, an acceptable internal consistency was found in 4 of them. In the case of the model with 4 factors, there was an acceptable internal consistency in 3 of them, which indicates that the results obtained from these factors are repeatable and consistent. Factors with unacceptable internal consistency suggest that the items are heterogeneous depending on what they claim to measure, since the omega is based on a factor analysis model that examines the interrelationships between elements and subsets of elements (Ferketich, 1990). Factor 2 increases its internal consistency in the model with 4 factors, which supports the idea of uniting factors 6 and 2 and demonstrates the relationships among emotional competence, interpersonal relationships and prosocial behaviors, as detected in a study by Pung et al. (2021). This study also pointed out that emotional competence and interpersonal relationships with peers promotes psychosocial behavior.
With respect to the Cronbach's α obtained in previous studies in the case of the model with 6 factors, a systematically lower value in factor 2 is notable [14, 17-19]. This again justifies the union of the factor related to the interpersonal relationships and prosocial behaviors in which the first acts as a mediator of the second; it is also based on the relationship between prosocial behaviors and variables existing in positive interpersonal behaviors such as empathy, desire for adventure, positive self-concept and self-esteem (Calvo et al., 2001).
Comment
The authors should check if common method biases exist when running measurement models.
Response:
This was a variable not considered at the beginning of the project. Although a more accurate test requires the joint application of the target instrument and another with an unrelated variable, with the available data, the Hartman factor test was performed. The results of performing an exploratory factor analysis using principal components and unweighted least squares, without rotating, yielded a 10-factor solution. The first explained 20% of the variance, and the first 5 together explained 42.76%. Although it is not irrefutable proof of the absence of common method biases, the consistency between the results obtained in the project presented with previous studies allows us to interpret that the data present a minor influence of this type of bias. Additionally, the presence of positive and negative items in the instrument used helps control response biases.
Reference: Hair, J., Black, W., Babin, B., Anderson, R. and Tatham, R. (2006). Multivariate Data Analysis. Upper Saddle River, NJ: Pearson Prentice-Hall.
Change:
No change.
Comment
Before running the SEM model, the authors should check outliers, normality etc. of the items used in the present study.
Response:
The suggestion was heeded, and the values obtained in the Anderson-Darlin and the Mardia skewness and Mardia kurtosis tests were noted.
Change:
3.2. Descriptive Analysis, Univariate and Multivariate Normality and Polychoric Correlation
Table 2 presents the descriptive analyses of the items of the PMHQ. There was no evidence of univariate (Anderson‒Darling test, minimum value = 22.19 and maximum value = 101.29, p < 0.01) or multivariate normality (Mardia Skewness = 24045.96; Mardia Kurtosis = 53.83, p < 0.01). In the Supplementary Materials Section, the polychoric correlation matrix of all 39 items is shown (Table 2A).
Comment
How did the authors convert the latent variables into observed variables? Factor score?
Response:
I'm not sure I understand this question. Perhaps it is related to the analysis of global internal consistency of an instrument that has different factors. If this is the case, the latter was removed from the manuscript. If you are referring to how instrument data is scored and interpreted, section 2.2.2 describes this information.
Change:
2.2.2. Positive Mental Health Questionnaire
The PMHQ was originally designed and validated by Lluch-Canut [24] in 1999 and later validated with a Mexican population of health sector workers by Aparicio et al. [41]. It is made up of six factors with a total of 39 items: personal satisfaction (PS; 8 items); prosocial attitude (PA; 5 items); self-control (SC; 5 items); autonomy (AU; 5 items); problem-solving and self-actualization (PSSA; 9 items); and interpersonal relationship skills (IR; 7 items). The instrument is answered on a 4-point Likert-type scale. For the positive items, the scale is always/almost always (4), frequently (3), sometimes (2) and never/almost never (1); for negative items, this scale is reversed. The positive items are 4, 5, 11, 15-18, 20-23, 25-29, 32, and 35-37. The negative items are 1-3, 6-10, 12-14, 19, 24, 30, 31, 33, 34, 38, and 39. In the validation carried out by Aparicio et al. [41], the percentage of variance explained was 43.4%, and Cronbach's α coefficient was 0.86 for the global instrument. These authors did not report the internal consistency obtained by factor. With the present population and prior to the psychometric analysis developed here, McDonald's omega (ω) coefficient per-factor were PS = 0.82; PA = 0.58; SC = 0.84; AU = 0.83; PSSA = 0.82; IR = 0.73).
Comment
Please remove Figure 1 and Figure 2, respectively.
Response:
The figures were removed. However, in the additional material, a table with the factor loadings was added.
Change:
Table 2B. Factorial loads per factor for each model
|
|
1 factor model |
4 factor model |
6 factors model |
||||||||
|
Item |
F1 |
F1 |
F2 |
F3 |
F4 |
F1 |
F2 |
F3 |
F4 |
F5 |
F6 |
|
R4 |
0.47 |
0.59 |
|
|
|
0.60 |
|
|
|
|
|
|
R6 |
0.45 |
0.56 |
|
|
|
0.56 |
|
|
|
|
|
|
R7 |
0.44 |
0.55 |
|
|
|
0.55 |
|
|
|
|
|
|
R12 |
0.66 |
0.81 |
|
|
|
0.82 |
|
|
|
|
|
|
R14 |
0.68 |
0.84 |
|
|
|
0.84 |
|
|
|
|
|
|
R31 |
0.65 |
0.81 |
|
|
|
0.81 |
|
|
|
|
|
|
R38 |
0.42 |
0.56 |
|
|
|
0.56 |
|
|
|
|
|
|
R39 |
0.47 |
0.62 |
|
|
|
0.63 |
|
|
|
|
|
|
R1 |
0.36 |
|
0.43 |
|
|
|
0.17 |
|
|
|
|
|
R3 |
0.23 |
|
0.33 |
|
|
|
0.35 |
|
|
|
|
|
R23 |
0.55 |
|
0.67 |
|
|
|
0.76 |
|
|
|
|
|
R25 |
0.38 |
|
0.50 |
|
|
|
0.59 |
|
|
|
|
|
R37 |
0.50 |
|
0.64 |
|
|
|
0.75 |
|
|
|
|
|
R8 |
0.38 |
|
0.49 |
|
|
|
|
0.53 |
|
|
|
|
R9 |
0.47 |
|
0.59 |
|
|
|
|
0.66 |
|
|
|
|
R11 |
0.34 |
|
0.45 |
|
|
|
|
0.83 |
|
|
|
|
R18 |
0.64 |
|
0.78 |
|
|
|
|
0.80 |
|
|
|
|
R20 |
0.39 |
|
0.48 |
|
|
|
|
0.79 |
|
|
|
|
R24 |
0.29 |
|
0.40 |
|
|
|
|
|
0.73 |
|
|
|
R30 |
0.39 |
|
0.46 |
|
|
|
|
|
0.66 |
|
|
|
R10 |
0.54 |
|
|
0.73 |
|
|
|
|
0.76 |
|
|
|
R13 |
0.48 |
|
|
0.66 |
|
|
|
|
0.55 |
|
|
|
R19 |
0.56 |
|
|
0.75 |
|
|
|
|
0.72 |
|
|
|
R33 |
0.43 |
|
|
0.55 |
|
|
|
|
|
0.12 |
|
|
R34 |
0.54 |
|
|
0.72 |
|
|
|
|
|
0.77 |
|
|
R15 |
0.51 |
|
|
|
0.53 |
|
|
|
|
0.63 |
|
|
R16 |
0.68 |
|
|
|
0.72 |
|
|
|
|
0.79 |
|
|
R17 |
0.56 |
|
|
|
0.60 |
|
|
|
|
0.74 |
|
|
R27 |
0.70 |
|
|
|
0.74 |
|
|
|
|
0.61 |
|
|
R28 |
0.66 |
|
|
|
0.70 |
|
|
|
|
0.71 |
|
|
R29 |
0.54 |
|
|
|
0.57 |
|
|
|
|
0.42 |
|
|
R32 |
0.62 |
|
|
|
0.66 |
|
|
|
|
0.75 |
|
|
R35 |
0.39 |
|
|
|
0.40 |
|
|
|
|
|
0.40 |
|
R36 |
0.66 |
|
|
|
0.70 |
|
|
|
|
|
0.61 |
|
R2 |
0.55 |
|
|
|
0.57 |
|
|
|
|
|
0.46 |
|
R5 |
0.59 |
|
|
|
0.62 |
|
|
|
|
|
0.81 |
|
R21 |
0.76 |
|
|
|
0.79 |
|
|
|
|
|
0.50 |
|
R22 |
0.73 |
|
|
|
0.76 |
|
|
|
|
|
0.41 |
|
R26 |
0.71 |
|
|
|
0.74 |
|
|
|
|
|
0.49 |
F: Factor
Reviewer 3 Report
Comments and Suggestions for Authors
In the manuscript, Dr. Luna and coauthors conducted validation study of diverse psychometric structures, aiming to evaluate the psychometric properties of the PMHQ using confirmatory factor analysis. In general, the manuscript is well-written in a good organization and in terms of grammar, with a very comprehensive and informative introduction, results and discussion. Methods are well-described, especially from statistical angle with comprehensive and robust justification. I have some minor concerns are as follows:
1. I felt that it is necessary to mention how many factors and items of PMHQ in the abstract, just giving audience a general impression of the psychometric dimension and how does the factor analysis work in terms of dimension reduction.
2. In Section 1.3, it is unclear for me to distinguish between several Postive Mental Health (PMH) measurements (i.e. PMHQ, PMH Scale, PMH-19, etc). I would like the authors to clarify the differences between those measurements.
3. In Section 1.5, I do not understand why there is a need to investigate the single factor analysis. It is hard to represent multiple domains (constructs) using single factor, while preserving major information. Reduction to three or four maybe more reasonable. Are there any previous studies working on the unifactorial analysis and prove the superiority over multifactorial ones?
4. I would like to see any visualization plots for missing data if possible (include them in separate appendix)
5. In Section 2.6.1, please add citations for Anderson-Darling test and Mardia’s coefficients.
6. I felt that it is necessary to include a comprehensive review of factor analysis in the Introduction part, as it is the primary statistical method used in this article
7. In Section 3.4.1, please justify why results were unstable from Table 4
8. For Figure 1, please kindly add footnotes to describe the figure (i.e. what is solid arrow, dashed arrow, etc)
I really appreciate the way your presented for the detailed Introduction, and the comprehensive and robust statistical methods used in this validation study.
Author Response
Comments and Suggestions for Authors
Comment
I felt that it is necessary to mention how many factors and items of PMHQ in the abstract, just giving audience a general impression of the psychometric dimension and how does the factor analysis work in terms of dimension reduction.
Response:
Mention was made in the abstract of the requested information
Change:
Abstract: The Positive Mental Health Questionnaire (PMHQ) has been validated across various populations but has displayed diverse psychometric structures depending on the procedures used. The original version of the PMHQ includes 39 items organized into 6 factors, although there are reports that indicate a reduced structure of between 1 and 4 factors. The aim of this study was to assess the psychometric properties of the PMHQ with 1, 4 and 6 factors. A total of 360 healthcare workers aged 23 to 77 (M = 37.06; SD = 10.79) participated. Construct validity was assessed through confirmatory factor analysis using weighted root mean square residual. The original 6-factor (χ2/df: 3.40; RMSEA: 0.085; CFI: 0.913; TLI: 0.906) and a reduced 4-factor (χ2/df: 2.90; RMSEA: 0.072; CFI: 0.931; TLI: 0.926) structure showed acceptable fit. The fit of the one-factor model was unacceptable. The internal consistency was evaluated through McDonald's ω, and it was acceptable for 4 of 6 factors of the original structure and for 3 of 4 factors of the reduced structure. In conclusion, these findings suggest that the 6-factor and 4-factor models are valid for measuring positive mental health. However, issues with internal consistency must be investigated.
Comment
In Section 1.3, it is unclear for me to distinguish between several Postive Mental Health (PMH) measurements (i.e., PMHQ, PMH Scale, PMH-19, etc.). I would like the authors to clarify the differences between those measurements.
Response:
The request was addressed, and more information was provided about the different instruments.
Change:
1.3. Self-report Instruments to Assess Positive Mental Health
The first instrument to be analyzed was the Lluch Positive Mental Health Questionnaire (PMHQ) [24], which we will discuss in detail later, since the present study sought its validation in healthcare workers in Mexico. The PMHQ has 39 items structured into 6 factors that correspond to the 6 positive mental health criteria of Jahoda [40], who conceives of mental health from an individual perspective. Although she accepts the impact of the environment and culture on health and illness, refuses to speak of “sick societies” because although it fully accepts the mutual influence between the physical and mental aspects of the human being, it does not consider that physical health is sufficient for good mental health. Another highly relevant instrument is the Mental Health Continuum-Large Form (MHC-LF) with 40 items, 7 focused on evaluating emotional well-being, 18 on psychological well-being, and 15 on social well-being, with adequate estimates of internal consistency. This instrument conceives a model that places health and illness as correlated unipolar dimensions that form a complete state of mental health, a single bipolar dimension, and that places mental health as the simple absence of psychopathology; likewise, the model on which it is based presumes that mental health is the ultimate goal of personal and social functioning [25]. Derived from the Mental Health Continuum-Large Form, the Mental Health Continuum-Short Form (MHC-SF) retains the 14 most prototypical items of the long version. Further, the MHC-SF points out that positive mental health is not simply the absence of mental illness but includes the presence of positive feelings (emotional well-being) and positive functioning in individual life (psychological well-being) and in community life (social well-being). This brief version assesses the frequency with which respondents experience positive mental health, with a score ranging from 0 (none of the time) to 5 (all the time), thus enabling their classification; for example, to have flourishing mental health they must experience every or nearly every day at least one of the three signs of hedonic well-being and at least six of the 11 signs of positive functioning during the past month. Regarding its psychometric properties, the MH-SF has shown excellent internal consistency and discriminant validity in adolescents and adults from the United States, the Netherlands, and South Africa [26–30]. Another relevant instrument is the General Health Questionnaire (GHQ-12), which is a screening instrument that detects nonpsychotic psychiatric disorders [31]; it has been translated and adapted into multiple languages, as it has the advantage of brevity and has shown adequate psychometric properties, which is why it is considered the most widely used screening instrument worldwide [32]. However, the disadvantage is a highly debated structure that some studies have indicated examines two factors: depression/anxiety and social dysfunction (Gureje, 1991; Kihç et al., 1997; Picardi et al., 2001; Politi et al., 1994; Schmitz et al., 1999; Werneke et al., 2000). Other studies describe three factors: coping strategies, self-esteem and stress (Campbell et al. 2003; French & Tait, 2004; Graetz, 1991; Sánchez-López et al., 2008), and there are authors who recommend using the GHQ-12 as a unidimensional screening instrument (Gao et al., 2004; Hankins, 2008; Rocha et al., 2011). Another instrument is the Positive Mental Health (PMH) instrument, which evaluates six factors: general coping, personal growth and autonomy, spirituality, interpersonal skills, emotional support and global affect; it consists of 47 items and was developed and validated with participants from China, Malaya, and India, showing high internal consistency and correlation with other measures of well-being [33]. A brief version was prepared from the PHQ of 47 items, and the 19-item Positive Mental Health (PMH-19), validated with populations from Singapore, China, Malaya, and India, also showed high internal consistency [34]. In addition, the Achutha Menon Centre Positive Mental Health Scale (AMCPMHS) developed for the Indian population is a valid and reliable instrument of 20 items organized into 4 dimensions: (1) realization of one’s own potential and belief in the dignity and worth of self, (2) utilization of coping abilities, (3) belief in the worth of others, and (4) work productivity and community contribution [35]. Lukat et al. [36] also developed a 9-item unidimensional positive mental health scale (PMH Scale) and validated it in various relevant groups. We can highlight the validation with parents of children with cancer, where it proved to be a valid, reliable and culturally relevant scale [37]. Finally, the Rapid Positive Mental Health Instrument (R-PMHI) is a 6-item unidimensional scale developed and validated with the Singaporean population [38]. In terms of validation with the Mexican population, we can highlight the Positive Mental Health Scale for Adults as relevant, which includes 7 factors and 83 items and has a global internal consistency α = 0.962 that ranges from.644 to.954 between factors [39].
Comment
In Section 1.5, I do not understand why there is a need to investigate the single factor analysis. It is hard to represent multiple domains (constructs) using single factor, while preserving major information. Reduction to three or four maybe more reasonable. Are there any previous studies working on the unifactorial analysis and prove the superiority over multifactorial ones?
Response:
Information was provided that aims to justify each research question and the objectives of the study.
Change:
Research Questions
The studies that have reported high correlations between PMHQ factors (e.g., [7,45,46,52]) have used exploratory factor analysis, which is a data-driven approach that lacks indices that allow evaluation of the goodness-of-fit of the models (Prokofieva et al., 2023). Confirmatory factor analysis, a theory-driven approach, can inflate the factor correlations and produce biased goodness-of-fit indices due to ignoring cross-loading (Alamer, 2022). When the correlation between factors ranges between 0.80 and 0.85, there is a risk of overfactoring the structure, which compromises its discriminant validity (Watkins, 2023). These facts motivated the first research question:
- Will the original structure of the PMHQ, based on previous studies (e.g., [7,45,46,52]) that reported high correlations between its factors, show poor fit?
Derived from the above, it is possible that a theory-and-data-driven modification of the PMHQ, based on the conceptual analysis of correlated factors, will allow the identification of a model with a better fit than the original structure. This motivated the second research question:
- Can a PMHQ structure with fewer than 6 factors (original structure) but more than 1 demonstrate an acceptable fit?
Finally, Cabarcas and Mendoza [42], through an exploratory factor analysis, detected that in the original 6-factor structure of the PMHQ, the first factor explained 67.18% of the variance. The remaining factors explained a value of less than 5%. This led them to evaluate a one-factor structure for this instrument. This motivated the third and last research question:
- Will a unifactorial structure of the PMHQ yield an acceptable fit?
In this context, the primary objective of this study was to evaluate the psychometric properties of the PMHQ from a structural equations approach, that is, by estimating its structure through CFA. A secondary objective was to evaluate the structure of this instrument with 1 factor or a smaller number of factors compared to the original structure but greater than 1.
Comment
I would like to see any visualization plots for missing data if possible (include them in separate appendix)
Response:
The missing data figure is presented. However, due to a comment from the first reviewer, a test was carried out to detect the structure of the missing data, and since it was not completely random, it was decided to perform a listwise deletion to retain only complete data.
Change:
Comment
In Section 2.6.1, please add citations for Anderson-Darling test and Mardia’s coefficients.
Response:
References were noted, and it was clarified that the evaluation of multivariate normality was analyzed using Mardia's skewness and kurtosis.
Change:
2.6.2. Descriptive Analysis, Univariate and Multivariate Normality and Polychoric Correlation
The mean and standard deviation of each item of the PMHQ and its coefficient of asymmetry and kurtosis were estimated. Univariate and multivariate normality was analyzed using the Anderson‒Darling test (Hawkins, 2023) and Mardia’s skewness and kurtosis (Wulandari et al., 2021), respectively. The report of multivariate outliers and multivariate normality supplements the scarce information about this variable in structural equation model studies (c.f., Lai & Zhang, 2017). The polychoric correlation of all 39 items of the PMHQ was estimated with the software FACTOR v.12.04.04 (Lorenzo-Seva & Ferrando, 2006; https://psico.fcep.urv.cat/utilitats/factor/index.html).
Comment
I felt that it is necessary to include a comprehensive review of factor analysis in the Introduction part, as it is the primary statistical method used in this article
Response:
A description of the confirmatory factor analysis was made, although it is brief due to the parsimony and fluidity of the manuscript. However, important aspects of this procedure were also noted as a conceptual justification of the research questions.
Change:
1.4. PMHQ Validation Studies
To evaluate positive mental health objectively and from a multifactorial model, Llinch developed the Positive Mental Health Questionnaire (PMHQ) based on the Jahoda model [40], which he validated through an exploratory factor analysis (EFA) among nursing students from the University of Barcelona. In this regard, factor analysis is a method that allows modeling the covariation between a set of observed variables based on a latent construct, and through EFA, the generation of hypotheses about its possible structure is pursued (Orçan, 2018); thus, in this exploration, Llinch detected the presence of six factors: personal satisfaction, prosocial attitude, self-control, autonomy, problem-solving and self-actualization and interpersonal relationship skills. Llinch obtained a global internal consistency α =.90 ranging from .58 to .82 between factors, with 46.8% of the variance explained [24]. After its development, favorable psychometric properties of the PMHQ have been found for health sector workers in Mexico [41], higher education students in Portugal [7] and Colombia [42], preprofessional psychology practitioners in Peru [43], children between 9 and 12 years of age from Mexico [44], nursing students from Catalonia, Spain [45], nursing university professors in Spain [46], Colombian youth between 13 and 25 years [47], and Colombian youth aged 12 and over from Arequipa, Peru [48].
However, the psychometric analyses of the PMHQ in the various populations have shown diverse psychometric structures, which is attributed to the diversity of procedures used. In the factor analysis (FA), for example, in the validations with workers from the health sector in Mexico, with university students from Portugal, with Mexican children and with young people between the ages of 13 and 25 in Colombia, the internal structure was evaluated using the principal components method [7,41,44,47], which is a nonrecommended factor extraction procedure [49] that ignores measurement error and tends to inflate factor loading and explained variance and overestimates dimensionality [50]. In the validation processes of the PMHQ, the type of correlation calculated between items prior to the FA was not reported when the polychoric correlations were the most appropriate given that the responses to the items were ordinal, which makes it improbable that the requirement of normal distributions was met [50]. The factor estimation method was also omitted in the reviewed studies, with the unweighted least squares (ULS) method being the most appropriate because it is the most robust and recommended in the event of a possible violation of the assumption of normality [49,50]. The differentiated factorial structure in the reports can also be the result of the rotation method used; in this regard, Aparicio et al. [41], Aguilar [43] and Gómez-Acosta et al. [47] chose an orthogonal rotation method, which assumes independence between factors without allowing us to detect the correlation between them. Finally, only a few studies [43–45,48] performed a confirmatory factor analysis (CFA) to evaluate the structure found in the EFA. In relation to this, importantly, the purpose of the CFA is to evaluate hypothetical structures of the latent constructs resulting from the EFA and/or develop a better understanding of said structures; this procedure is a specific form of structural equation (Orçan, 2018) in which researchers present a prespecified factor solution, which is evaluated in terms of how well it reproduces the sample covariance matrix of the measured variables (Brown & Moore, 2012).
In most of the reported studies, internal consistency was obtained through the alpha coefficient, except the validation carried out in Peru with participants older than 12 years, in which internal consistency was obtained through the omega coefficient. In this regard, it has been shown that the alpha coefficient has limitations, being susceptible to the number of items on the scale, the number of response options and the proportion of variance of the test [51].
1.5. Research Questions
The studies that have reported high correlations between PMHQ factors (e.g., [7,45,46,52]) have used exploratory factor analysis, which is a data-driven approach that lacks indices that allow evaluation of the goodness-of-fit of the models (Prokofieva et al., 2023). Confirmatory factor analysis, a theory-driven approach, can inflate the factor correlations and produce biased goodness-of-fit indices due to ignoring cross-loading (Alamer, 2022). When the correlation between factors ranges between 0.80 and 0.85, there is a risk of overfactoring the structure, which compromises its discriminant validity (Watkins, 2023). These facts motivated the first research question:
- Will the original structure of the PMHQ, based on previous studies (e.g., [7,45,46,52]) that reported high correlations between its factors, show poor fit?
Derived from the above, it is possible that a theory-and-data-driven modification of the PMHQ, based on the conceptual analysis of correlated factors, will allow the identification of a model with a better fit than the original structure. This motivated the second research question:
- Can a PMHQ structure with fewer than 6 factors (original structure) but more than 1 demonstrate an acceptable fit?
Finally, Cabarcas and Mendoza [42], through an exploratory factor analysis, detected that in the original 6-factor structure of the PMHQ, the first factor explained 67.18% of the variance. The remaining factors explained a value of less than 5%. This led them to evaluate a one-factor structure for this instrument. This motivated the third and last research question:
- Will a unifactorial structure of the PMHQ yield an acceptable fit?
In this context, the primary objective of this study was to evaluate the psychometric properties of the PMHQ from a structural equations approach, that is, by estimating its structure through CFA. A secondary objective was to evaluate the structure of this instrument with 1 factor or a smaller number of factors compared to the original structure but greater than 1.
Comment
In Section 3.4.1, please justify why results were unstable from Table 4
Response:
This problem was solved by constraining some values of six model factors.
Change:
3.3.1. Original Structure with 6 Factors
Confirmatory factor analysis of the original version with 6 factors of the PMHQ did not show a positive definite covariance matrix. Table 3 (without constraints) shows the correlation matrix between the six factors and reveals a high correlation for Factor 3 with Factor 5 and for Factor 2 and Factor 6. Additionally, an eigenvalue was negative (i.e., -0.0007). A covariation constraint for both pairs of factors was fixed at 0.05, and the model was estimated again. The covariance matrix was now positive definite with all eigenvalues > 0. The correlation matrix between factors is shown in Table 3 (with constraints). All factor loadings were significant (p < 0.001). The standardized λ ranged between 0.12 and 0.84 (see Table 2B in Supplementary Materials Section), and all covariances between factors were significant (range for standardized values = 0.16 to 0.79, p < 0.05). The goodness-of-fit indices and mean factor loading for this structure are shown in Table 4. Except for the χ2 test and WRMR, the rest were acceptable. All Mλ were < 0.70.
Table 3. Correlation between factors of the Positive Mental Health Questionnaire with 6 factors
|
|
Without constraint |
With constraint |
||||||||
|
Factors |
F1 |
F2 |
F3 |
F4 |
F5 |
F1 |
F2 |
F3 |
F4 |
F5 |
|
F2 |
0.34 |
|
|
|
|
0.31 |
|
|
|
|
|
F3 |
0.60 |
0.53 |
|
|
|
0.60 |
0.49 |
|
|
|
|
F4 |
0.65 |
0.22 |
0.61 |
|
|
0.65 |
0.16 |
0.61 |
|
|
|
F5 |
0.53 |
0.79 |
0.83 |
0.52 |
|
0.53 |
0.79 |
0.79 |
0.50 |
|
|
F6 |
0.61 |
0.88 |
0.59 |
0.47 |
0.67 |
0.61 |
0.73 |
0.59 |
0.47 |
0.67 |
Abbreviations: F: Factor.
Table 4. Goodness-of-fit indices and mean factor loading for the structures evaluated with the Positive Mental Health Questionnaire
|
Factors |
χ2 |
χ2/df |
RMSEA (90% CI) |
WRMR |
CFI |
TLI |
Mλ |
|||||
|
F1 |
F2 |
F3 |
F4 |
F5 |
F6 |
|||||||
|
F6 |
2346.27** |
3.40 |
0.085 (0.081 to 0.089) |
1.65 |
0.913 |
0.906 |
0.67 |
0.52 |
0.72 |
0.68 |
0.61 |
0.52 |
|
F4 |
2018.81** |
2.90 |
0.075 (0.072 to 0.079) |
1.53 |
0.931 |
0.926 |
0.67 |
0.52 |
0.68 |
0.65 |
||
|
F1 |
2898.64** |
4.12 |
0.097 (0.093 to 0.100) |
1.86 |
0.885 |
0.878 |
0.52 |
|||||
Abbreviations: F: Factor. Note: ** p ≤ 0.001.
Comment
For Figure 1, please kindly add footnotes to describe the figure (i.e. what is solid arrow, dashed arrow, etc)
Response:
At the request of Reviewer 2, the figures were removed. However, in the additional material, a table with the factor loadings was added.
Change:
Table 2B. Factorial loads per factor for each model
|
|
1 factor model |
4 factor model |
6 factors model |
||||||||
|
Item |
F1 |
F1 |
F2 |
F3 |
F4 |
F1 |
F2 |
F3 |
F4 |
F5 |
F6 |
|
R4 |
0.47 |
0.59 |
|
|
|
0.60 |
|
|
|
|
|
|
R6 |
0.45 |
0.56 |
|
|
|
0.56 |
|
|
|
|
|
|
R7 |
0.44 |
0.55 |
|
|
|
0.55 |
|
|
|
|
|
|
R12 |
0.66 |
0.81 |
|
|
|
0.82 |
|
|
|
|
|
|
R14 |
0.68 |
0.84 |
|
|
|
0.84 |
|
|
|
|
|
|
R31 |
0.65 |
0.81 |
|
|
|
0.81 |
|
|
|
|
|
|
R38 |
0.42 |
0.56 |
|
|
|
0.56 |
|
|
|
|
|
|
R39 |
0.47 |
0.62 |
|
|
|
0.63 |
|
|
|
|
|
|
R1 |
0.36 |
|
0.43 |
|
|
|
0.17 |
|
|
|
|
|
R3 |
0.23 |
|
0.33 |
|
|
|
0.35 |
|
|
|
|
|
R23 |
0.55 |
|
0.67 |
|
|
|
0.76 |
|
|
|
|
|
R25 |
0.38 |
|
0.50 |
|
|
|
0.59 |
|
|
|
|
|
R37 |
0.50 |
|
0.64 |
|
|
|
0.75 |
|
|
|
|
|
R8 |
0.38 |
|
0.49 |
|
|
|
|
0.53 |
|
|
|
|
R9 |
0.47 |
|
0.59 |
|
|
|
|
0.66 |
|
|
|
|
R11 |
0.34 |
|
0.45 |
|
|
|
|
0.83 |
|
|
|
|
R18 |
0.64 |
|
0.78 |
|
|
|
|
0.80 |
|
|
|
|
R20 |
0.39 |
|
0.48 |
|
|
|
|
0.79 |
|
|
|
|
R24 |
0.29 |
|
0.40 |
|
|
|
|
|
0.73 |
|
|
|
R30 |
0.39 |
|
0.46 |
|
|
|
|
|
0.66 |
|
|
|
R10 |
0.54 |
|
|
0.73 |
|
|
|
|
0.76 |
|
|
|
R13 |
0.48 |
|
|
0.66 |
|
|
|
|
0.55 |
|
|
|
R19 |
0.56 |
|
|
0.75 |
|
|
|
|
0.72 |
|
|
|
R33 |
0.43 |
|
|
0.55 |
|
|
|
|
|
0.12 |
|
|
R34 |
0.54 |
|
|
0.72 |
|
|
|
|
|
0.77 |
|
|
R15 |
0.51 |
|
|
|
0.53 |
|
|
|
|
0.63 |
|
|
R16 |
0.68 |
|
|
|
0.72 |
|
|
|
|
0.79 |
|
|
R17 |
0.56 |
|
|
|
0.60 |
|
|
|
|
0.74 |
|
|
R27 |
0.70 |
|
|
|
0.74 |
|
|
|
|
0.61 |
|
|
R28 |
0.66 |
|
|
|
0.70 |
|
|
|
|
0.71 |
|
|
R29 |
0.54 |
|
|
|
0.57 |
|
|
|
|
0.42 |
|
|
R32 |
0.62 |
|
|
|
0.66 |
|
|
|
|
0.75 |
|
|
R35 |
0.39 |
|
|
|
0.40 |
|
|
|
|
|
0.40 |
|
R36 |
0.66 |
|
|
|
0.70 |
|
|
|
|
|
0.61 |
|
R2 |
0.55 |
|
|
|
0.57 |
|
|
|
|
|
0.46 |
|
R5 |
0.59 |
|
|
|
0.62 |
|
|
|
|
|
0.81 |
|
R21 |
0.76 |
|
|
|
0.79 |
|
|
|
|
|
0.50 |
|
R22 |
0.73 |
|
|
|
0.76 |
|
|
|
|
|
0.41 |
|
R26 |
0.71 |
|
|
|
0.74 |
|
|
|
|
|
0.49 |
F: Factor
Round 2
Reviewer 1 Report
Comments and Suggestions for Authors
no further comments
Author Response
Thank you very much for your review and contributions to the final version of the manuscript.
Reviewer 2 Report
Comments and Suggestions for Authors
Review Comments
10-11-2023
1. The author used lavaan package to do the CFA analysis, however, the specific version of the package and R (Rstudio) should be transparently disclosed to the readers for reproduction of the analysis.
2. There are many issues in Table 1. More specifically, the 1st column should be “Variable” rather than “Variables”. The variable names below the first column should be moved down by 3 rows each, to align with the specific entries of the corresponding categories. You can look at where this thing goes. In addition, why do the authors use the phrase “civic status”? It should be “Martial Status”. Moreover, what does “area” mean? It should be “Study Area”. Lastly, the decimal should be aligned in the last column.
3. The authors should identify the method used to handle missing data and address the assumptions underlying the method. More specifically, Little’s MCAR test should be used in the present study to achieve the goal mentioned. In addition, the proportions of missing data across the variables in the study should be provided.
4. It's precisely because the author used Little's MCAR test to confirm the pattern of missing values in the research sample as belonging to MCAR, allowing for their removal. So, the analytic samples should not include missing values. Please remove the rows in Table 1 that contain the information about missing values.
5. Subtitle 2.6 “Data Analysis” is incorrect. It should be “Data Analytic Strategy”.
6. The content in line 348 should be removed.
7. The paragraph from line 349 and 356 should be moved to the subsection of “Participants”.
8. The relations between the factors and the observed variable should be clearly presented when
articulating the model(s) under study. Up until now, we have only discussed models that specify factors as the causal agents of the observed variables. In such models, the observed variables are hypothesized to be correlated with one another because they are a function of the same factor. Given this conceptualization of the factor–variable relation, the factor is deemed latent and the
direct paths flow from the latent variable to the observed variables. It is also possible to conceptualize factor analysis of a factor as being a function of the observed variables (e.g., overall stress is a function of work stress, spouse-related stress, and children-related stress). The direct paths flow from the observed variables to the emergent factor. Anyway, should clearly present how the variables are related to the factor and explain why the particular model chosen (latent or emergent factor) is appropriate.
9.For the reliability of the scale used in the present study, the authors did some work. However, it is not sufficient. Item reliability, scale reliability, and factor reliability. If reliability is adequate, external validity evidence should be gathered. The quality of the factor is ultimately dictated by how well observed relations with other constructs align with theoretical expectations.
10. Table 2 is quite messy. First, the name of the 1st column should be “Item No.”. Second, In the second column, all text descriptions below 'Item Content' should be left-aligned, except for the centered content 'Factor 1, Factor 2, Factor 3, and Factor 4. The authors should present the original 'Item Content' in a concise manner, without the need for direct copying, while retaining its original meaning within the limited space available. Moreover, the decimal in the last column should be aligned.
11. This is an incorrect manner to add notes below a Table-Abbreviations: K: kurtosis; M: mean; S: skewness; SD: standard deviation. (Line 364)
12. The subtitle from line 317 to 318 should be changed to “Descriptive Statistics of the Items”.
13. All “p” denoting “p-value” in the manuscript should be italicized.
14. All “Abbreviations” in the manuscript should be replaced with “Notes:”.
15. p<=0.001 in line 383 should be removed, which is meaningless.
16. Note all Tables in the manuscript should just have 3 lines. Please adjust the tables incompatible with this requirement.
Comments on the Quality of English Language
Please improve English writing skills.
Author Response
Comments and Suggestions for Authors
- The author used lavaan package to do the CFA analysis, however, the specific version of the package and R (Rstudio) should be transparently disclosed to the readers for reproduction of the analysis.
Response:
- For each library used, the version was noted, since there is no way to justify noting only the version of one of them and not the others.
Change:
The data from the PMHQ were analyzed using the R v.4.3.1 program [79] and its RStudio v.2023.06.1 interface [80]. The packages dplyr v.1.1.2 [81], mice v.3.16.0 [82], naniar v.1.0.0 [83], psych v.2.3.9 [84], MVN v.5.9 [85], stats v.4.3.1 [79], PerformanceAnalytics v.2.0.4 [86], lavaan v.0.6-16 [87], semPlot v.1.1.6 [88] and misty v.0.5.3 [89] were used.
- There are many issues in Table 1. More specifically, the 1stcolumn should be “Variable” rather than “Variables”. The variable names below the first column should be moved down by 3 rows each, to align with the specific entries of the corresponding categories. You can look at where this thing goes. In addition, why do the authors use the phrase “civic status”? It should be “Marital Status”. Moreover, what does “area” mean? It should be “Study Area”. Lastly, the decimal should be aligned in the last column.
Response:
- Each of the noted observations was addressed. However, it was not possible to move the name of the variables 3 columns since in some cases that would leave said labels outside the Table.
Change:
|
Variable |
Category |
n (%) |
|
Sex |
Female |
273 (75.8) |
|
|
Male |
86 (23.9) |
|
|
Missing |
1 (0.3) |
|
Marital Status |
Single |
217 (60.3) |
|
|
Married |
86 (23.9) |
|
|
Consensual union |
40 (11.1) |
|
|
Divorced |
9 (2.5) |
|
|
Separated |
3 (0.8) |
|
|
Widowed |
5 (1.4) |
|
Educational level |
Technical school |
14 (3.9) |
|
|
Bachelor’s degree |
121 (33.6) |
|
|
Specialty |
110 (30.6) |
|
|
Subspecialty |
89 (24.7) |
|
|
Master’s degree |
22 (6.1) |
|
|
PhD |
3 (0.8) |
|
|
Missing |
1 (0.3) |
|
Study Area |
Medical |
237 (65.8) |
|
|
Nursing |
123 (34.2) |
|
Shift |
Morning |
178 (49.4) |
|
|
Evening |
25 (6.9) |
|
|
Night |
30 (8.3) |
|
|
Other |
127 (35.3) |
|
Second job in public sector |
Yes |
31 (8.6) |
|
|
No |
329 (91.4) |
|
Second job in private practice |
Yes |
49 (13.6) |
|
|
No |
311 (86.4) |
|
Diagnosis of psychopathology in the past 12 months |
Yes |
62 (17.2) |
|
|
No |
298 (82.8) |
|
Use of psychotropic drugs in the past 12 months |
Yes |
68 (18.9) |
|
|
No |
292 (81.1) |
- The authors should identify the method used to handle missing data and address the assumptions underlying the method. More specifically, Little’s MCAR test should be used in the present study to achieve the goal mentioned. In addition, the proportions of missing data across the variables in the study should be provided.
Response:
- The effect of presenting Missing not at random (MNAR) data was briefly noted, as was the effect of removing it. Considering that the total amount of missing data is 0.02% of the data and that it corresponds to 39 variables, presenting the proportion of missing data per variable is not very relevant. Therefore, the request stating: “the proportions of missing data across the variables in the study should be provided” was not attended to. However, the reviewer was provided with a graph with their structure, which is presented in Figure S1A in the Supplementary Materials. Additionally, in Section 3.1. Data Processing indicates that the origin (unobserved variable) of these missing data was the application of the printed questionnaire.
Change:
The presence of missing data was evaluated. Little's [90] missing completely at random (MCAR) test was used to detect their structure. In the case of missing data that were not missing completely at random, a listwise deletion was employed. This was done to avoid incorporating bias into the data analysis. However, the unobserved data that could have caused said missing data was analyzed. With a database free of missing data, the presence of multivariate outliers was analyzed using the Mahalanobis distance (D2; [91]) with a cutoff point established by the χ2 value with degrees of freedom equal to the number of items in the PMHQ (i.e., [25]) and a p < 0.001, although these were not withdrawn unless they revealed consistent patterns of mechanical response (c.f., [92]).
3.1. Data Processing
In a database with 360 observations for 39 variables, 34 missing data points were found, corresponding to 0.2% of the total (see Figure S1A in the Supplementary Materials). There were no missing data completely at random, according to Little´s test (χ2 = 782, p = 0.03); therefore, each observation with at least one missing data point was removed from the dataset. This eliminated 24 observations coming from the collection of information made with the printed questionnaire and left n = 336 for posterior analysis. After this, 22 multivariate outliers were identified. However, visual inspection of the cases did not reveal the presence of systematic patterns of mechanical response, so the observations were kept for further analysis.
- It's precisely because the author used Little's MCAR test to confirm the pattern of missing values in the research sample as belonging to MCAR, allowing for their removal. So, the analytic samples should not include missing values. Please remove the rows in Table 1 that contain the information about missing values.
Response:
- The observation is appreciated. However, the authors consider not addressing it. It seems to us that the reviewer confuses the collected sample with the analytic sample. The sociodemographic data of the total participants (n = 360) contain a missing variable in the case of the sex of one of them. Regarding the total of this sample, the presence of missing data was analyzed to carry out the data analysis and the presence of 24 incomplete observations was discovered, which were eliminated to leave n = 336 that was subject to factor analysis. Presenting it in this way is useful to make the presence of missing data transparent and at the same time make explicit that the data analyzes were done on complete data.
- Subtitle 2.6 “Data Analysis” is incorrect. It should be “Data Analytic Strategy”.
Response:
- The observation was attended to.
Change:
2.6. Data Analytic Strategy
The data from the PMHQ were analyzed using the R v.4.3.1 program [79] and its RStudio v.2023.06.1 interface [80]. The packages dplyr v.1.1.2 [81], mice v.3.16.0 [82], naniar v.1.0.0 [83], psych v.2.3.9 [84], MVN v.5.9 [85], stats v.4.3.1 [79], PerformanceAnalytics v.2.0.4 [86], lavaan v.0.6-16 [87], semPlot v.1.1.6 [88] and misty v.0.5.3 [89] were used.
- The content in line 348 should be removed.
- The paragraph from line 349 and 356 should be moved to the subsection of “Participants”.
Response:
- The observation is appreciated. However, the authors consider not addressing it. Present in different sections the sampling strategy and the characteristics of the recruited participants (Section 2.1. Participants), the preparation strategy of the data collected from the sample prior to analysis (Section 2.6.1. Data Processing) and the results of said analysis. preparation (Section 3.1. Data Processing) help make this entire process explicit in a structured way.
- The relations between the factors and the observed variable should be clearly presented when
articulating the model(s) under study. Up until now, we have only discussed models that specify factors as the causal agents of the observed variables. In such models, the observed variables are hypothesized to be correlated with one another because they are a function of the same factor. Given this conceptualization of the factor–variable relation, the factor is deemed latent and the
direct paths flow from the latent variable to the observed variables. It is also possible to conceptualize factor analysis of a factor as being a function of the observed variables (e.g., overall stress is a function of work stress, spouse-related stress, and children-related stress). The direct paths flow from the observed variables to the emergent factor. Anyway, should clearly present how the variables are related to the factor and explain why the particular model chosen (latent or emergent factor) is appropriate.
Response:
- The reviewer's comment points to a conceptual issue about the way in which the factors of the evaluated instrument are formed. However, it is not clear enough to indicate an express request for Change or the incorporation of new information or interpretation. Additionally, the objective of the study was to evaluate different structures of the target instrument. The original structure, derived from a previously reported theory and data. A 4-factor structure, proposed through a data and theory-driven strategy, made explicit in the manuscript itself. And finally a 1-factor structure, based on previous data (i.e., data-driven). That objective was achieved and the results interpreted conceptually and empirically. The comment that the reviewer points out could be considered for a study in the planning phase in which the instrument evaluated in this work will be analyzed again, but from a different approach. Specifically with exploratory structural equation models.
9.For the reliability of the scale used in the present study, the authors did some work. However, it is not sufficient. Item reliability, scale reliability, and factor reliability. If reliability is adequate, external validity evidence should be gathered. The quality of the factor is ultimately dictated by how well observed relations with other constructs align with theoretical expectations.
Response:
- The comment is noted and for future studies it is planned to evaluate the concurrent and divergent validity of the instrument used. The latter was not the objective of the present study, so there is no data to evaluate this type of validity.
- Table 2 is quite messy. First, the name of the 1st column should be “Item No.”. Second, In the second column, all text descriptions below 'Item Content' should be left-aligned, except for the centered content 'Factor 1, Factor 2, Factor 3, and Factor 4. The authors should present the original 'Item Content' in a concise manner, without the need for direct copying, while retaining its original meaning within the limited space available. Moreover, the decimal in the last column should be aligned.
Response:
- The observation was addressed except for the request: “The authors should present the original 'Item Content' in a concise manner, without the need for direct copying, while retaining its original meaning within the limited space available.”. The authors consider that the best way to retain the original concept of an item in an instrument is to present the item as it was written in said instrument.
Change:
|
Item No. |
Item content |
M |
SD |
S |
K |
|
|
Factor 1: Personal satisfaction |
|
|
|
|
|
4 |
I like myself as I am. |
3.51 |
0.72 |
-1.40 |
1.40 |
|
6 |
I feel like I am about to explode. |
3.30 |
0.72 |
-1.00 |
1.21 |
|
7 |
I find life to be boring and monotonous. |
3.61 |
0.68 |
-1.94 |
3.75 |
|
12 |
I view the future with pessimism. |
3.76 |
0.57 |
-2.70 |
7.94 |
|
14 |
I see myself as less important than those around me. |
3.76 |
0.59 |
-2.85 |
8.23 |
|
31 |
I feel inept and useless. |
3.85 |
0.51 |
-4.01 |
16.89 |
|
38 |
I feel unsatisfied with myself. |
3.40 |
0.98 |
-1.52 |
1.01 |
|
39 |
I feel unsatisfied with the way I look. |
3.38 |
0.82 |
-1.29 |
1.05 |
|
|
Factor 2: Prosocial attitude |
|
|
|
|
|
1 |
I find it especially difficult to accept others when their attitudes are different from mine. |
3.40 |
0.67 |
-1.02 |
1.31 |
|
3 |
I find it particularly difficult to listen to people tell me their problems. |
3.65 |
0.64 |
-2.16 |
5.25 |
|
23 |
I feel that I am someone to be trusted. |
3.86 |
0.45 |
-3.79 |
16.59 |
|
25 |
I consider the needs of others. |
3.23 |
0.84 |
-0.71 |
-0.54 |
|
37 |
I like to help others. |
3.65 |
0.62 |
-1.87 |
3.43 |
|
|
Factor 3: Self-control |
|
|
|
|
|
2 |
Problems often cause me to feel blocked. |
3.44 |
0.66 |
-1.20 |
2.01 |
|
5 |
I am able to control myself when I feel negative emotions. |
3.12 |
0.82 |
-0.52 |
-0.57 |
|
21 |
I am able to control myself when I have negative thoughts. |
3.28 |
0.80 |
-0.76 |
-0.41 |
|
22 |
I am able to maintain a high level of self-control in conflictive situations in my life. |
3.33 |
0.71 |
-0.62 |
-0.66 |
|
26 |
When I experience unpleasant external pressure, I am able to maintain my personal balance. |
3.24 |
0.77 |
-0.79 |
0.14 |
|
|
Factor 4: Autonomy |
|
|
|
|
|
10 |
I worry a lot about what others think of me. |
3.32 |
0.77 |
-1.04 |
0.75 |
|
13 |
The opinions of others have a strong influence on me when I have to make decisions. |
3.49 |
0.67 |
-1.24 |
1.46 |
|
19 |
It troubles me when people criticize me. |
3.31 |
0.79 |
-0.96 |
0.28 |
|
33 |
I find it hard to hold my own opinions. |
3.57 |
0.71 |
-1.67 |
2.29 |
|
34 |
When I have to make big decisions, I feel very unsure of myself. |
3.34 |
0.79 |
-1.25 |
1.31 |
|
|
Factor 5: Problem-solving and self-actualization |
|
|
|
|
|
15 |
I am able to make decisions on my own. |
3.68 |
0.69 |
-2.35 |
5.13 |
|
16 |
I try to look for the positive side when bad things happen to me. |
3.45 |
0.69 |
-0.96 |
0.10 |
|
17 |
I try to improve myself as a person. |
3.73 |
0.59 |
-2.27 |
4.92 |
|
27 |
When there are changes in my surroundings, I try to adapt to them. |
3.60 |
0.63 |
-1.43 |
1.49 |
|
28 |
In the face of a problem, I am able to ask for information. |
3.59 |
0.66 |
-1.52 |
1.71 |
|
29 |
I find changes in my daily routine to be stimulating. |
3.23 |
0.76 |
-0.52 |
-0.71 |
|
32 |
I try to develop my abilities to the maximum. |
3.61 |
0.64 |
-1.73 |
3.03 |
|
35 |
I am able to say no when I want to. |
3.15 |
0.92 |
-0.69 |
-0.67 |
|
36 |
When I am faced with a problem, I try to find possible solutions. |
3.73 |
0.54 |
-2.22 |
5.62 |
|
|
Factor 6: Interpersonal relationship skills |
|
|
|
|
|
8 |
I find it particularly difficult to provide emotional support to others. |
3.48 |
0.71 |
-1.39 |
1.85 |
|
9 |
I find it hard to establish deep and satisfying interpersonal relationships with some people. |
3.50 |
0.71 |
-1.35 |
1.42 |
|
11 |
I feel that I have a strong ability to put myself in the shoes of others and to understand their responses. |
3.04 |
0.93 |
-0.57 |
-0.73 |
|
18 |
I consider myself to be a good professional. |
3.74 |
0.54 |
-2.34 |
6.25 |
|
20 |
I think that I am a sociable person. |
3.19 |
0.84 |
-0.67 |
-0.52 |
|
24 |
I find it particularly hard to understand the feelings of others. |
3.40 |
0.73 |
-1.10 |
0.88 |
|
30 |
I find it hard to relate openly with my teachers/bosses. |
3.35 |
0.83 |
-1.22 |
0.84 |
- This is an incorrect manner to add notes below a Table-Abbreviations: K: kurtosis; M: mean; S: skewness; SD: standard deviation. (Line 364)
- All “Abbreviations” in the manuscript should be replaced with “Notes:”.
Response:
- The observation was attended to.
Change:
Notes: K: kurtosis; M: mean; S: skewness; SD: standard deviation.
Notes: F: Factor.
Notes: F: Factor. Note: ** p < 0.05.
Notes: F: Factor.
Notes: α: Cronbach’s alpha; ω: McDonald’s omega.
- The subtitle from line 317 to 318 should be changed to “Descriptive Statistics of the Items”.
Response:
- The observation was attended to.
Change:
2.6.2. Descriptive Statistics of the Items
- All “p” denoting “p-value” in the manuscript should be italicized.
Response:
- The observation was attended to.
Change:
and a p < 0.001, although these were not withdrawn
Items with p < 0.05 were retained
value = 101.29, p < 0.01) or multivariate normality (Mardia skewness = 24045.96; Mardia kurtosis = 53.83, p< 0.01)
factor loadings were significant (p < 0.001).
values = 0.16 to 0.79, p < 0.05).
Notes: F: Factor. Note: ** p < 0.05.
All factor loadings were significant (p < 0.001).
(range for standardized values = 0.39 to 0.65, p < 0.001).
All factor loadings were significant (p < 0.001).
- p<=0.001 in line 383 should be removed, which is meaningless.
Response:
- The observation was attended to and replaced by p < 0.05.
Change:
Notes: F: Factor. Note: ** p < 0.05.
- Note all Tables in the manuscript should just have 3 lines. Please adjust the tables incompatible with this requirement.
Response:
- The observation was attended to.
Change:
|
|
Without constraint |
With constraint |
||||||||
|
Factors |
F1 |
F2 |
F3 |
F4 |
F5 |
F1 |
F2 |
F3 |
F4 |
F5 |
|
F2 |
0.34 |
|
|
|
|
0.31 |
|
|
|
|
|
F3 |
0.60 |
0.53 |
|
|
|
0.60 |
0.49 |
|
|
|
|
F4 |
0.65 |
0.22 |
0.61 |
|
|
0.65 |
0.16 |
0.61 |
|
|
|
F5 |
0.53 |
0.79 |
0.83 |
0.52 |
|
0.53 |
0.79 |
0.79 |
0.50 |
|
|
F6 |
0.61 |
0.88 |
0.59 |
0.47 |
0.67 |
0.61 |
0.73 |
0.59 |
0.47 |
0.67 |
|
|
F1 |
F2 |
F3 |
|
F2 |
0.53 |
|
|
|
F3 |
0.65 |
0.39 |
|
|
F4 |
0.59 |
0.70 |
0.58 |
|
Factors |
F1 |
F2 |
F3 |
F4 |
F5 |
F6 |
|
6 |
α = 0.75; ω = 0.82 |
α = 0.47; ω = 0.58 |
α = 0.78; ω = 0.84 |
α = 0.69; ω = 0.83 |
α = 0.77; ω = 0.82 |
α = 0.59; ω = 0.73 |
|
4 |
α = 0.75; ω = 0.82 |
α = 0.70; ω = 0.75 |
α = 0.85; ω = 0.87 |
α = 0.69; ω = 0.83 |
|
|
|
1 |
α = 0.89; ω = 0.90 |
|
|
|
|
|